

# Systematics, distribution patterns and historical biogeography of the Central America wandering spider genus *Kiekie* Polotow & Brescovit, 2018 (Araneae: Ctenidae)

Nicolas Hazzi[1,2,3] and Gustavo Hormiga[2]

[1] National Museum of Natural History, Smithsonian Institution, Washington, DC, United States
[2] Biological Sciences, The George Washington University, Washington, DC, United States
[3] Fundacion Ecotonos, Cali, Colombia

Corresponding author
Nicolas Hazzi,
nicolashazzi@hotmail.com

## ABSTRACT

*Kiekie* Polotow & Brescovit, 2018 is a Neotropical genus of Ctenidae, with most of its species occuring in Central America. In this study, we review the systematics of *Kiekie* and describe five new species and the unknown females of *K. barrocolorado* Polotow & Brescovit, 2018 and *K. garifuna* Polotow & Brescovit, 2018, and the unknown male of *K. verbena* Polotow & Brescovit, 2018. In addition, we described the female of *K. montanense* which was wrongly assigned as *K. griswoldi* Polotow & Brescovit, 2018 (both species are sympatric). We provided a modified diagnosis for previously described species based on the morphology of the newly discovered species and *in situ* photographs of living specimens. We inferred a molecular phylogeny using four nuclear (histone H3, 28S rRNA, 18S rRNA and ITS-2) and three mitochondrial genes (cytochrome c oxidase subunit I or COI, 12S rRNA and 16S rRNA) to test the monophyly of the genus and the evolutionary relationships of its species. Lastly, we reconstruct the historical biogeography and map diversity and endemism distributional patterns of the different species. This study increased the number of known species of *Kiekie* from 13 to 18, and we describe a new genus, *Eldivo* which is sister lineage of *Kiekie*. Most of the diversity and endemism of the genus *Kiekie* is located in the montane ecosystems of Costa Rica followed by the lowland rainforest of the Pacific side (Limon Basin). *Kiekie* originated in the North America Tropical region, this genus started diversifying in the Late Miocene and spread to Lower Central America and South America. In that region, *Kiekie* colonized independently several times the montane ecosystems corresponding to periods of uplifting of Talamanca and Central Cordilleras.

## INTRODUCTION

The family Ctenidae includes medium to large (5–50 mm) wandering spiders that typically inhabit the forest floor and low vegetation, although a few species are arboreal (*Gasnier et al., 2009*). To date, more than 600 species in 48 genera have been recorded in this family,

most of them distributed in the tropics (*World Spider Catalog, 2022*). Their large size and predominant abundance in most Neotropical rainforests suggest that ctenids play an important role in tropical ecosystems as top generalist predators of invertebrates and small vertebrates (*Folt & Lapinski, 2017*). Recently, *Polotow & Brescovit (2018)* described the genus *Kiekie* to accommodate two species of the polyphyletic *Ctenus* and other nine new species. Recently, *Omelko (2023)* described two new additional species to the genus. *Kiekie* reaches its higher species diversity in Central America with 13 species distributed (mainly in Costa Rica) and one species in South America (*Polotow & Brescovit, 2018*; *Omelko, 2023*). Some species of *Kiekie* are restricted to primary forests and their abundances decrease in less pristine habitats (*Hazzi et al., 2020*). In addition, *Kiekie* species are restricted to different biomes, encompassing high montane to lowland areas, and dry to rain forests biomes.

Lower Central America is a highly diverse region that provides a geologically complex and dynamic model for studying the assembly and spatial evolution of a Neotropical biota (*Bagley & Johnson, 2017*; *Mendoza et al., 2019*). Three important tectonic events have been suggested to summarize the main changes in the structural styles along the forearc of this region: (1) subduction of an ancient ridge during the Neogene (*Brandes & Winsemann, 2018*); (2) reorganization of tectonic plates during the middle to late Miocene (*Mescua et al., 2017*; *Porras et al., 2021*); and arrival of the Cocos Ridge at the Middle America trench in the Pliocene (*Abratis & Wöorner, 2001*; *Morell, 2016*). The high diversity of *Kiekie* in this region, and the restricted distribution of most of its species to specific ecosystems and elevation gradients, makes this genus a good model to study how geological events have contributed to the diversification of the Lower Central American fauna.

While carrying extensive fieldwork in Central America and having examined collections in several natural history museums from Central America, we have discovered several new species of *Kiekie* and new geographic range extensions for the majority of previously described species. In addition, we were able to get fresh samples of most of the *Kiekie* species suitable for genetic sequencing and phylogenetic analysis. Therefore, in this study we review the systematics of *Kiekie* and described six new species and the unknown females of *K. barrocolorado Polotow & Brescovit, 2018* and *K. garifuna Polotow & Brescovit, 2018*, and the unknown male of *K. verbena*. In addition, we described the female of *K. montanense Polotow & Brescovit, 2018* (*Polotow & Brescovit (2018)* was incorrectly assigned as the female of *K. griswoldi Polotow & Brescovit, 2018* as the female of *K. montanense*- these two species are sympatric). We inferred a molecular phylogeny using four nuclear (histone H3, 28S, 18S and ITS-2) and three mitochondrial genes (12S, 16S and COI) to test the monophyly of *Kiekie* and the evolutionary relationships of its species. With this framework in place, we address the following questions relating to the historical biogeography of the genus: 1) Does the higher diversity of *Kiekie* in Central America correspond with the ancestral area of the genus? or did *Kiekie* originate in South America and dispersed to Central America? and 2) Did *Kiekie* originate in lowland or in montane ecosystems? What is the evolutionary history of transitions between lowland and montane habitats? Shortly after the completion of our study two new species of *Kiekie* from Panama

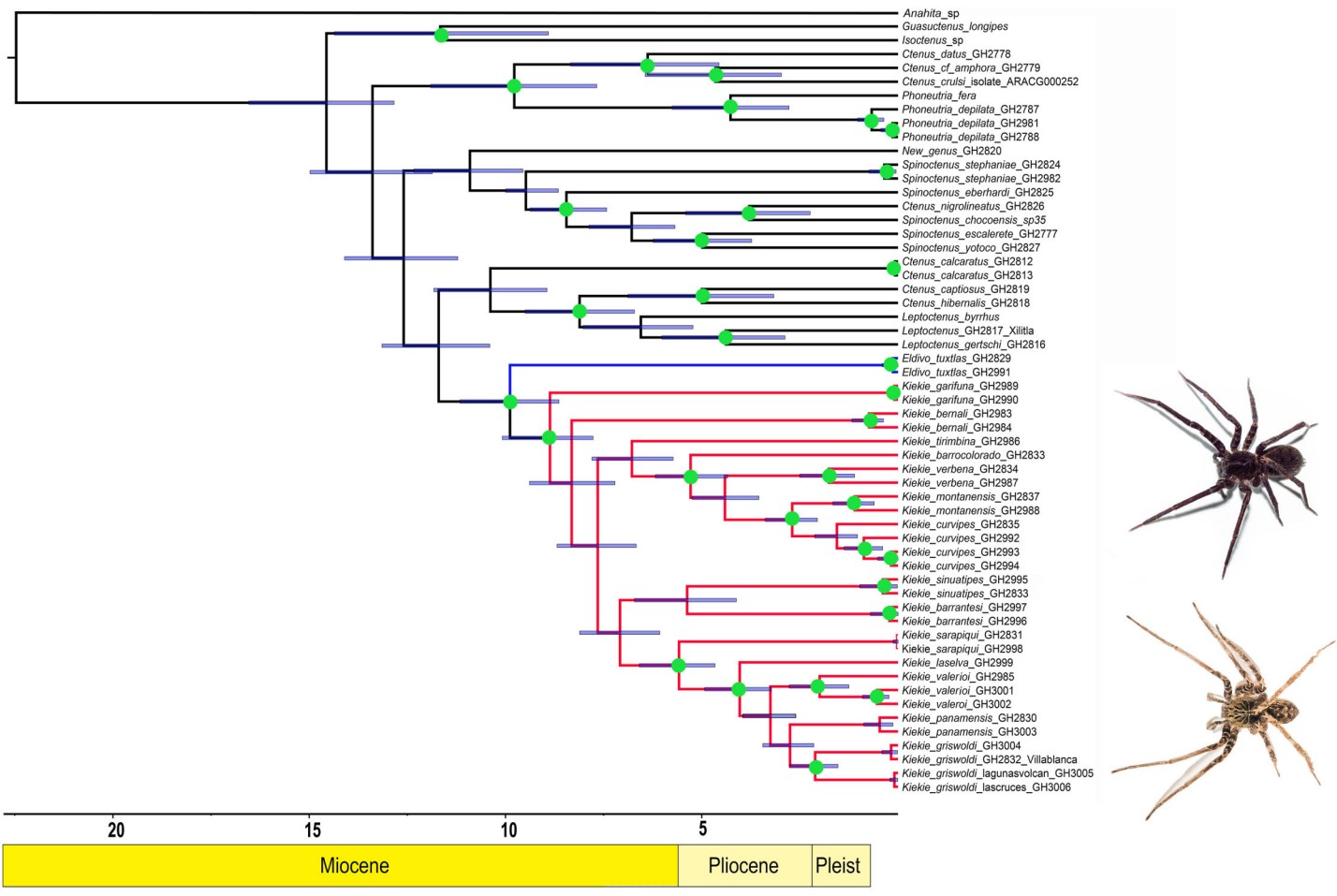

**Figure 1 Likelihood tree.** Likelihood tree estimated in IQTREE and time calibrated in BEAST depicting the divergence times and phylogenetic relationships of *Kiekie* and Cteninae outgroups. Bars represent the 95% highest posterior density interval. Green circles at nodes indicated ultrafast bootstrap support values above 95. Blue clade: *Eldivo*, red clade: *Kiekie*. Photos credit: Alejandra Arroyave.

were described by *Omelko (2023)* and these are now included in the taxonomic section with a revised generic diagnosis. Although we have not examined the holotypes of these two latter species, the descriptions and diagnoses are sufficient for accurate identifications. In the present work we have increased the number of described species of *Kiekie* from 13 to 18 and discovered a new genus, *Eldivo* gen. n., which is sister lineage of *Kiekie*. The figures of this work is organized in the following order: phylogenetic analysis (Fig. 1), distribution patterns and biogeography (Figs. 2 and 3), species descriptions in alphabetic order (Figs. 4–26), and species distribution maps (Fig. 27).

## MATERIALS AND METHODS

### Morphological examination and description of species

Specimens were preserved in 95% ethanol. Descriptions and terminology follow *Höfer, Brescovit & Gasnier (1994)* and *Simó & Brescovit (2001)*. All measurements were given in

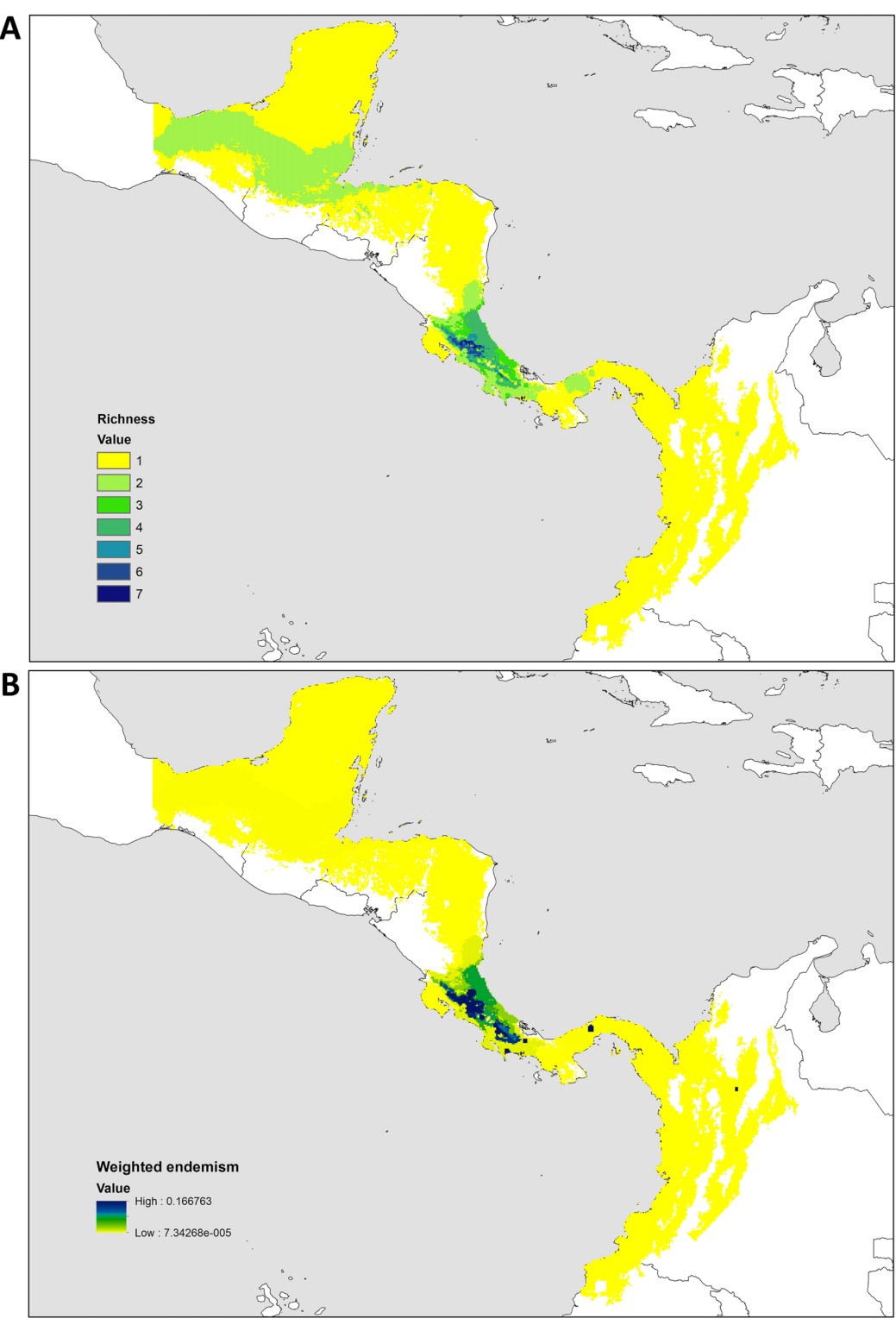

**Figure 2** **Distribution patterns of** *Kiekie*. (A) Species richness; (B) weighted endemism.

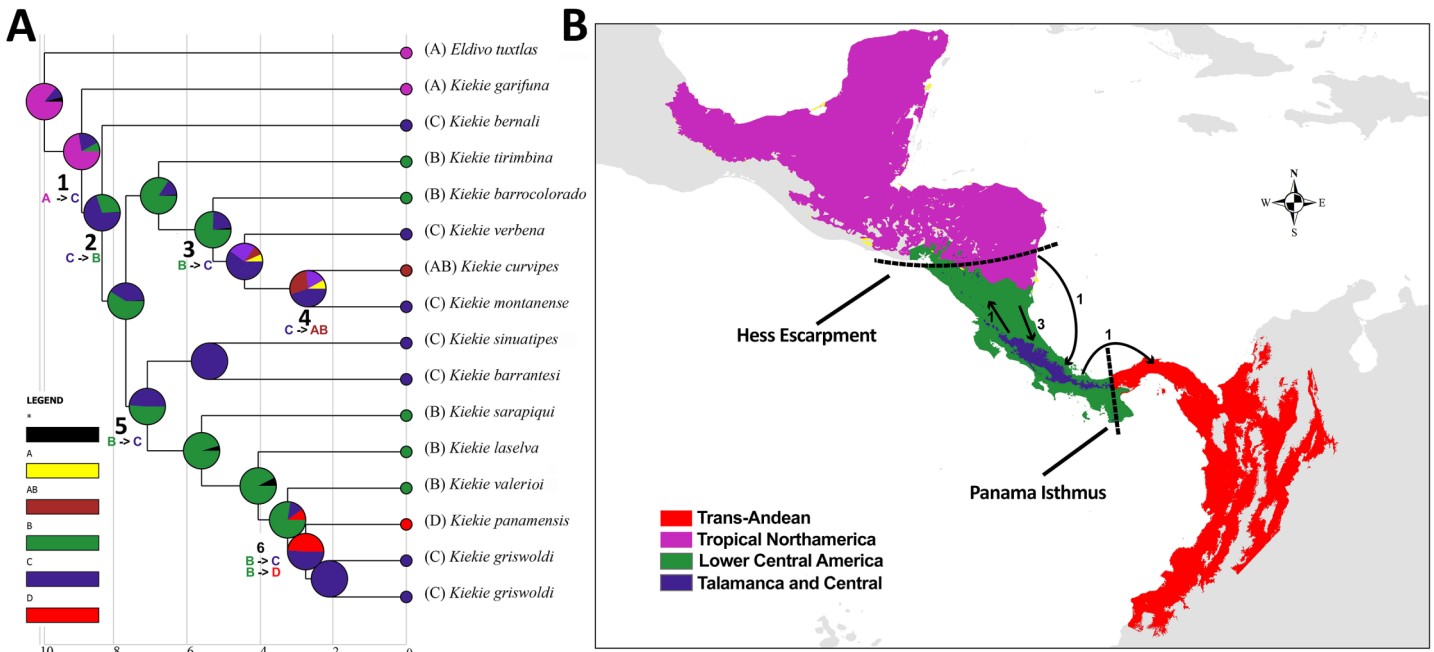

**Figure 3 Historical Biogeography.** (A) Ancestral area distribution reconstructed by DEC+J model in RASP. Pie charts at the nodes give the relative frequencies of the ancestral-area reconstruction. The numbers in the tree correspond with dispersal events described in the results. (B) Map of biogeographic areas choose for ancestral biogeographic reconstruction depicting main geographical barriers in Central America and also the number of dispersal events between them.

millimeters and were taken using Leica Application Software (3.4.2) in a Leica M205A stereomircroscope and a Leica DFC425 camera. Epigyna were digested with pancreatin solution (*Álvarez-Padilla & Hormiga, 2007*) to enable the study of internal structures. Digital images were made with a Leica M205A stereomicroscope adapted to a Leica DFC425 camera digital camera. Extended focal range images were composed using the software package Helicon Focus (version 6.7.1; www.heliconsoft.com) from Helicon Soft Ltd. Digital SEM photographs were taken using a LEO 1430VP scanning electron microscope from the scanning laboratory, Department of Biology of The George Washington University. For SEM preparation, structures were cleaned ultrasonically, transferred to 95% (if they were in 75%–80% ethanol originally) and then to 100% ethanol for 10 min in each immersion before being air-dried. The specimens were then coated with Au-Pd. The following abbreviations are used: C, conductor; CD, copulatory ducts; CO, copulatory opening; E, embolus; FD, fertilization ducts; LP, lateral projection; MA, median apophysis; MF, median field of epigynum; RTA, retrolateral tibial apophysis; S, spermathecae. The distribution maps were elaborated with ArcGIS (ESRI) v 10, with the spatial analyst extension.

## DNA SEQUENCING METHODS

Collecting in Costa Rica and exporting the specimens out the country was permitted by Sistema Nacional de Areas de Conservacion (SINAC) and Ministerio de Ambiente de Energia (SINAC-ACC-PI-R-045-2019 and PE-DCUSBSE-SE325-2019). Collecting in

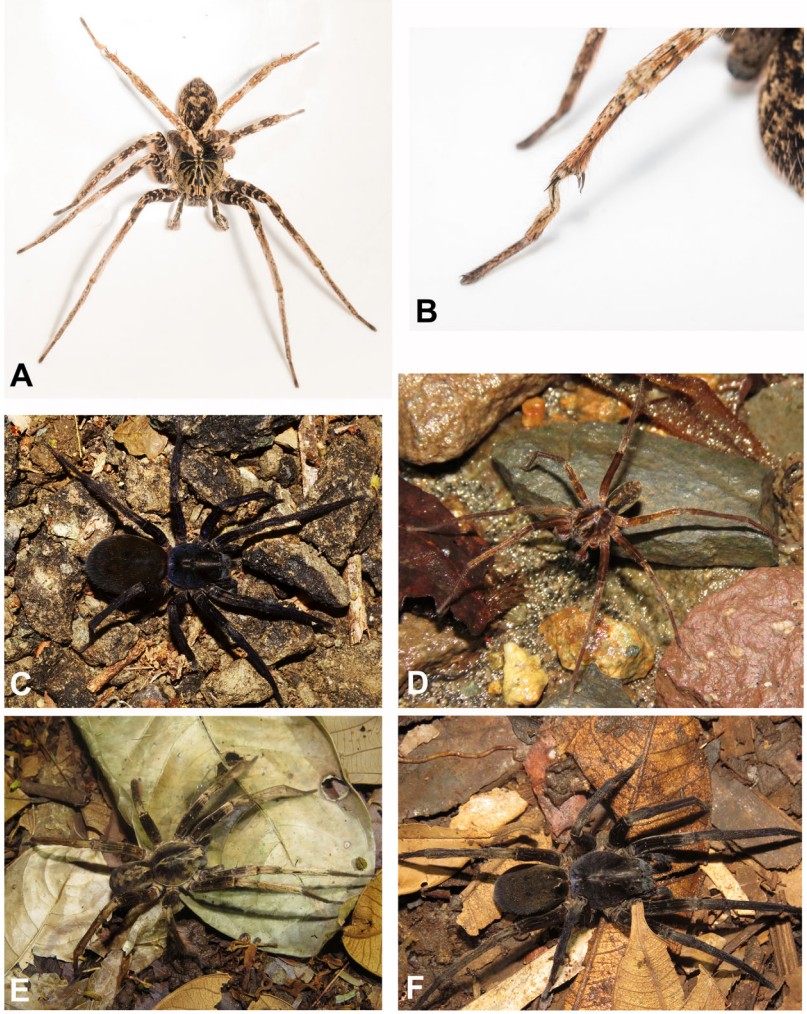

**Figure 4 Habitus of *Kiekie* spp.** (A) Male of *Kiekie panamensis*; (B) modified male metatarsus IV of *K. panamensis*; (C) female of *K. panamensis*, (D) male of *K. bernali*; (E) male of *K. laselva*; (F) female of *K. laselva*.

Panama and exporting the specimens out the country was permitted by Ministerio de Ambiente (SEX/A-79-18). *Sampling design.* We sequenced a total of 30 *Kiekie* specimens representing 14 of the 16 known species. When possible, each species was represented by a male and a female, in order to ensure the correct sex was matched for each species. Based on the phylogenetic relationships among genera within Cteninae (*Hazzi & Hormiga, 2023*), we included taxa representing the "American clade" as an outgroup, and rooted the tree with *Anahita* sp. The final matrix includes 57 terminals, of which 30 belong to *Kiekie* (Table S1). DNA was extracted from the coxal and femoral tissues utilizing the Qiagen DNEasy kit, while the remainder of the sample was retained for voucher purposes.

The analysis incorporated seven genetic markers: two mitochondrial ribosomal markers, 12S rRNA (~400 bp) and 16S rRNA (~550 bp); three cytoplasmic ribosomal markers, 18S rRNA (~800 bp), 28S rRNA (~2,200 bp), and ITS-2 (~350 bp); a nuclear protein-coding gene, histone H3 (~320 bp); and a mitochondrial protein-coding gene, cytochrome c

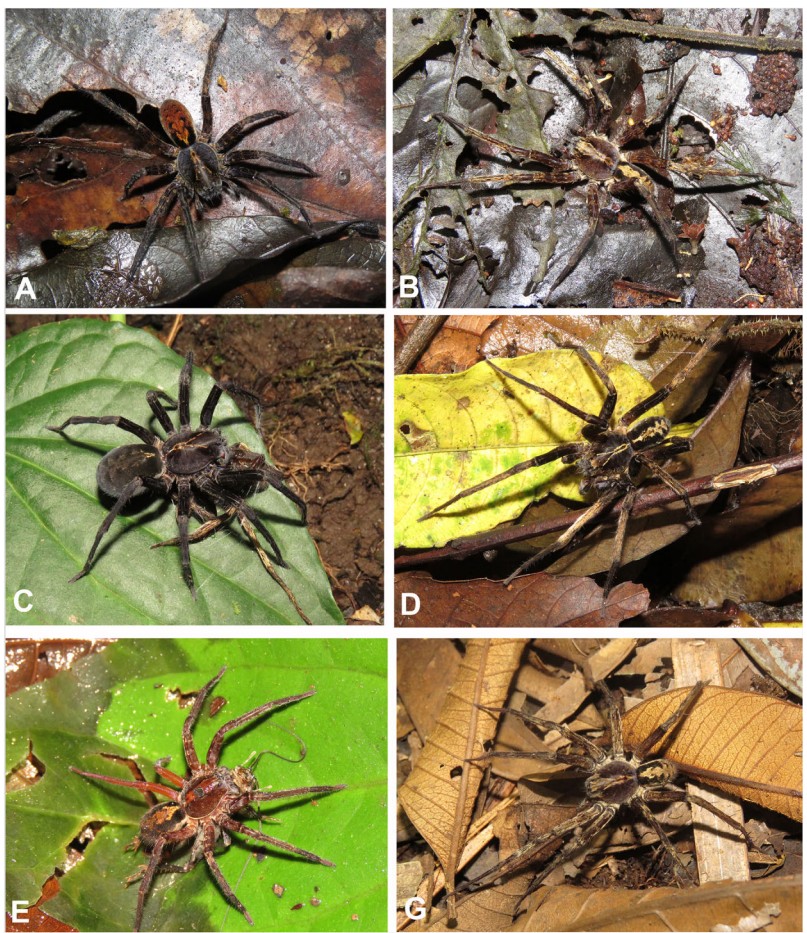

**Figure 5 Habitus of *Kiekie* spp.** (A) Female of *Kiekie montanensis*; (B) male of *K. montanensis*; (C) female of *K. griswoldi*; (D) male of *K. griswoldi*; (E) female of *K. curvipes*; (F) male of *K. curvipes*.

oxidase subunit I (COI) (~800 bp). Amplification was performed using the Promega GoTaq kit and the primers detailed in *Hazzi & Hormiga (2023)*. Contigs were assembled with GENEIOUS 6.0.6 (http://www.geneious.com; *Kearse et al., 2012*), and sequences of protein-coding genes were examined for stop codons. These sequences were also compared with the NCBI BLAST nucleotide database to ensure there was no contamination.

## Phylogenetic analysis

Phylogenetic analyses were performed using equal weight parsimony (MP) and maximum likelihood (ML). For the model-based analysis, the best partitioning scheme and substitution models were explored using ModelFinder implemented in IQTREE (*Nguyen et al., 2015*; *Kalyaanamoorthy et al., 2017*), selecting partition merging and the corrected Akaike information criterion (AICc). We partitioned protein coding genes (H3 and COI) into codon position, and each ribosomal gene was treated as a whole (ITS-2, 28S, 12S, 16S and 18S), for a total of eleven partitions. The maximum likelihood analyses were performed with the package IQ-TREE 1.4.2 (*Nguyen et al., 2015*) and ultrafast bootstrap

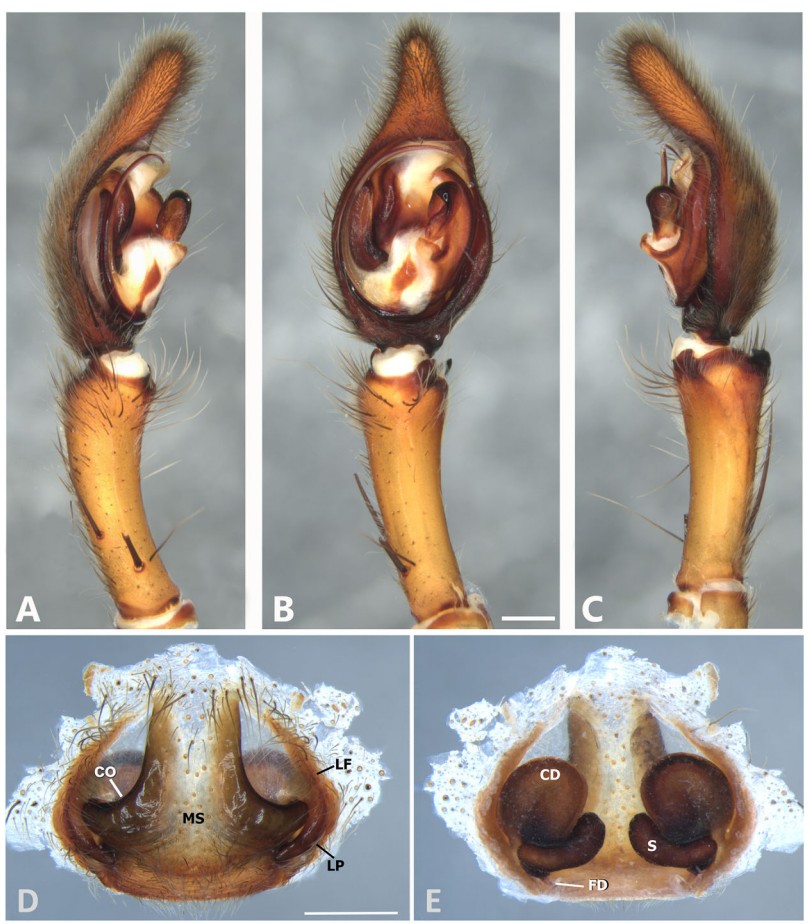

**Figure 6 _Kiekie barrantesi_ sp. nov, left male palp.** (A) Ventral view; (B) prolateral view; (C) retrolateral view; scale bar = 2.00 mm; (D) epigynum, ventral view; (E) vulva, dorsal view. CD, copulatory duct; CO, copulatory opening; FD, fertilization duct; LF, epigynal lateral field; MS, epigynal median sector; S, spermatheca. Scale bars = 1.00 mm.                                

(_Minh, Nguyen & Von Haeseler, 2013_) were used as a support measure. The parsimony analyses were carried out in TNT v. 1.5 (_Goloboff & Catalano, 2016_) using 500 random addition sequences followed by TBR branch swapping algorithm and retaining 10 trees per replicate until the length was hit five times. Branch support was assessed using 1,000 replicates of jackknife resampling (_Farris et al., 1996_).

We conducted a dated phylogeny estimation using Bayesian inference analysis through BEAST version 2.5.2 (_Bouckaert et al., 2014_).The analysis was carried out with an uncorrelated lognormal clock and and a birth–death model for the tree prior. All markers were unlinked for site and clock models apart from the mitochondrial rRNA genes 16S and 12S because ModelFinder suggested treating the mitochondrial rRNA genes 16S and 12S as a combined partition due to their similarities; hence, they were not unlinked for site and clock models. Conversely, all other markers were treated independently for both site and clock models. To reduce computational time and address issues with chain convergence in the dating analysis, the tree topology derived from likelihood estimation was transformed into an ultrametric tree. This transformation was achieved by employing a penalized

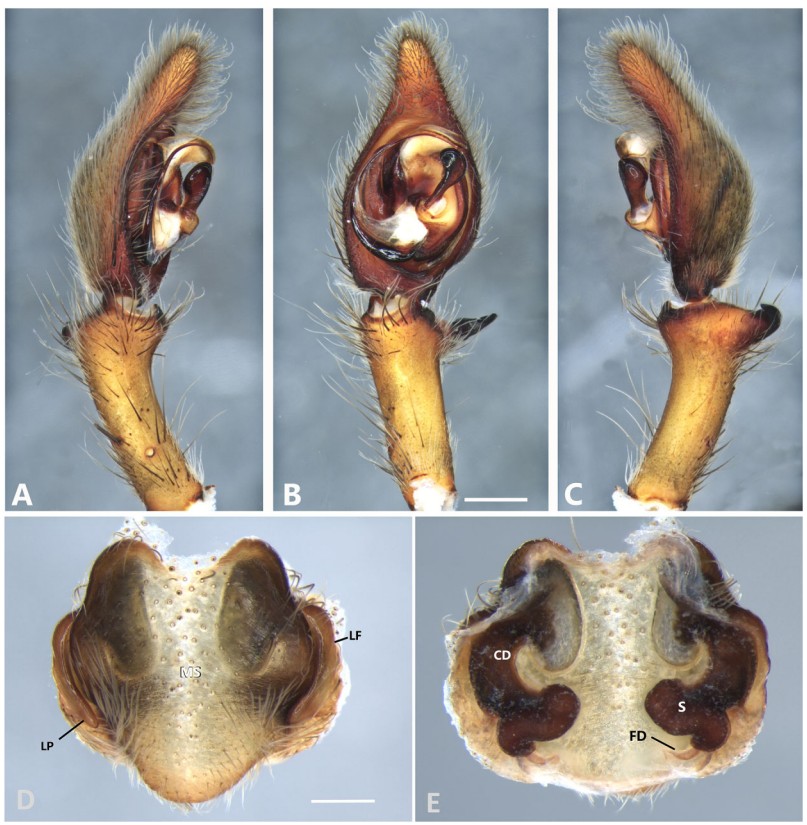

**Figure 7** *Kiekie barrocolorado* **sp. nov, left male palp.** (A) Ventral view; (B) prolateral view; (C) ret­rolateral view, arrow indicates small peak of the median apophysis; scale bar = 1.00 mm; (D) epigynum, ventral view; (E) vulva, dorsal view. CD, copulatory duct; FD, fertilization duct; LF, epigynal lateral field; MS, epigynal median sector; S, spermatheca. Scale bar = 0.50 mm.

likelihood approach using the 'chronos' function from the R package 'ape' version 5.3 (*Paradis & Schliep, 2019*). We executed three separate analyses, each consisting of 200 million generations across four Markov Chain Monte Carlo (MCMC) chains. The results, post the initial burn-in phase, were consolidated using Logcombiner version 2.5.1. Sampling of trees and parameters occurred at intervals of every 10,000 generations, with the initial 25% of samples being discarded as burn-in. The remaining samples were then utilized to compute the posterior parameters. Chain convergence was evaluated using Tracer version 1.7 (*Rambaut et al., 2018*), ensuring that the Effective Sample Size (ESS) exceeded 200. The summarization of trees was performed using TreeAnnotator, included in the BEAST package.

Due to the lack of fossils in Ctenidae, we used gene substitution rates reported for lycosoid spiders (*Piacentini & Ramírez, 2019*). We prefer to use these rates rather than the rates reported in *Hazzi & Hormiga (2023)* because of the lower rates reported in mitochondrial genes due to genome saturation. In addition, we incorporated one calibration point within Ctenidae based on a biogeographic event related to biome formation in the Tropical Andes (*Jaramillo, 2019*). We set a maximum crown age of 10 Ma

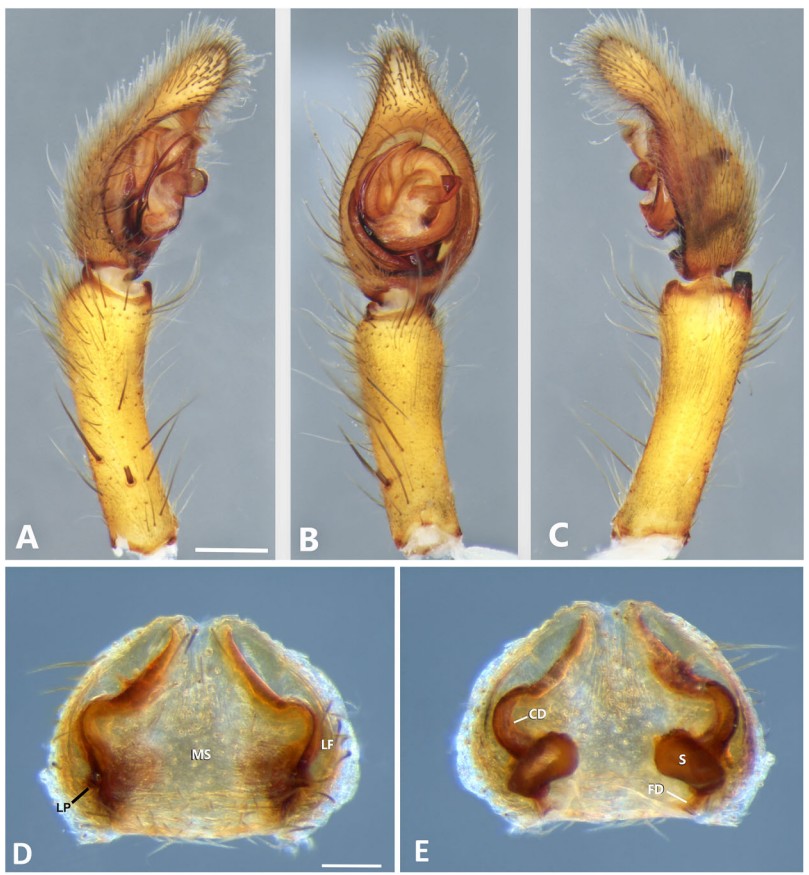

**Figure 8** ***Kiekie bernali* sp. nov, left male palp.** (A) Ventral view; (B) prolateral view; (C) retrolateral view; scale bar = 0.50 mm; (D) epigynum, ventral view; (E) vulva, dorsal view. CD, copulatory duct; FD, fertilization duct; LF, epigynal lateral field; MS, epigynal median sector; S, spermatheca. Scale bars = 0.20 mm.               

with a uniform distribution for *Spinoctenus Hazzi et al., 2018*. The genus *Spinoctenus* is composed of 12 species distributed in the tropical Andes and Chocó biogeographical regions (*Hazzi et al., 2018*). Most *Spinoctenus* diversity is found in the Andes (eight species) and using event-based biogeographic analyses on a morphological phylogeny, *Hazzi et al. (2018)* inferred an Andean origin for this genus. Both the most recent molecular phylogeny of Ctenidae (*Hazzi & Hormiga, 2023*) and the morphological phylogeny of *Hazzi et al. (2018)* indicate that the Andean species *Spinoctenus eberhardi Hazzi et al., 2018* and *S. stephaniae Hazzi et al., 2018* diverged early within the genus. These two species are endemic to cloud forest ecosystems above 2,000 m, a biome that was completely absent from the north of South America before 10 Ma (*Jaramillo, 2019*).

## Species model distributions and distribution patterns of diversity and endemism

We estimated the distribution of *Kiekie* species using the Maxent algorithm (*Phillips, Anderson & Schapire, 2006*; *Elith et al., 2011*). Occurrence data was compiled from published sources, direct field observations, and museum collections. To address potential

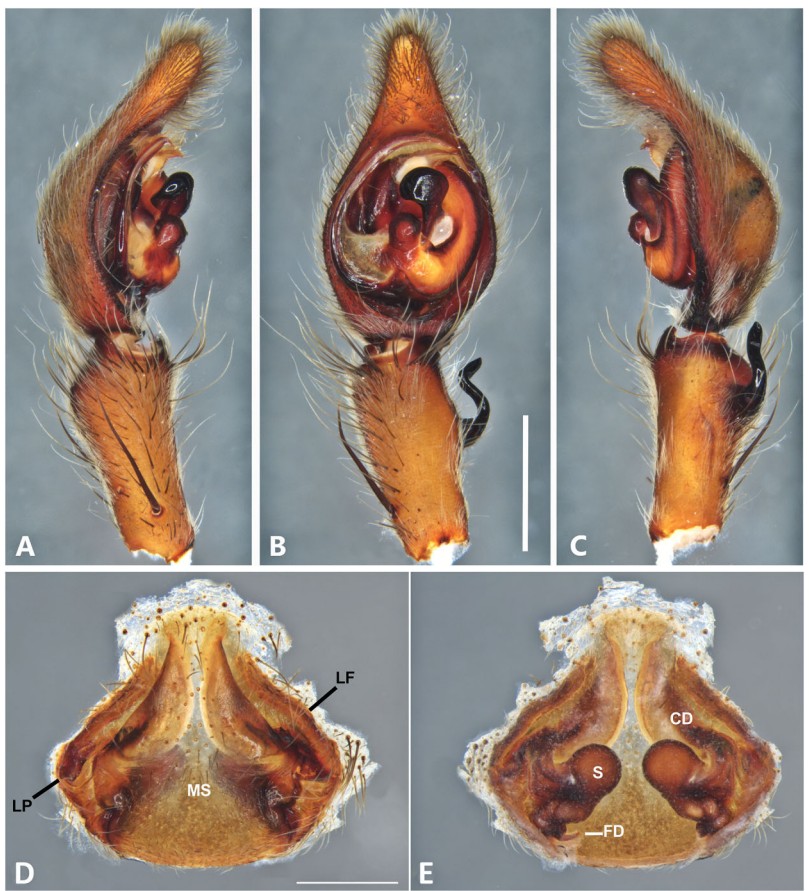

**Figure 9** ***Kiekie curvipes* from Tirimbina Biological Station (Costa Rica), left male palp.** (A) Ventral view; (B) prolateral view; (C) retrolateral view, arrow indicates small peak of the median apophysis; scale bar = 2.00 mm; (D) epigynum, ventral view; (E) vulva, dorsal view. CD, copulatory duct; FD, fertilization duct; LF, epigynal lateral field; MS, epigynal median sector; S, spermatheca. Scale bar = 1.00 mm.

biases from disproportionate sampling, we implemented a spatial filtering method, retaining only a single record within every 10 km radius. We sourced 19 bioclimatic layers that summarize annual patterns, seasonal variations, and extreme weather conditions related to precipitation and temperature from the WorldClim database at a spatial resolution of 30 arc-seconds, equivalent to 1 km² (*Fick & Hijmans, 2017*). To minimize predictor variable collinearity, we chose variables with a Pearson correlation coefficient of less than 0.7, including annual mean temperature (Bio1), mean diurnal range (Bio2), temperature seasonality (Bio4), annual precipitation (Bio12), precipitation seasonality (Bio15), and precipitation of the warmest quarter (Bio18). The area selected for modelling was the trans-Andean region through the Neotropical Caribbean area of Mexico.

We ran the models selecting a maximum number of background points of 10,000 and random seed, and choosing logistic function output. Model performance was evaluated by employing a k-fold cross-validation method, which partitioned the occurrence data into a 70% portion for training and a 30% segment for testing. This division was conducted 15 times to ensure robustness in the validation process. he performance of the models was

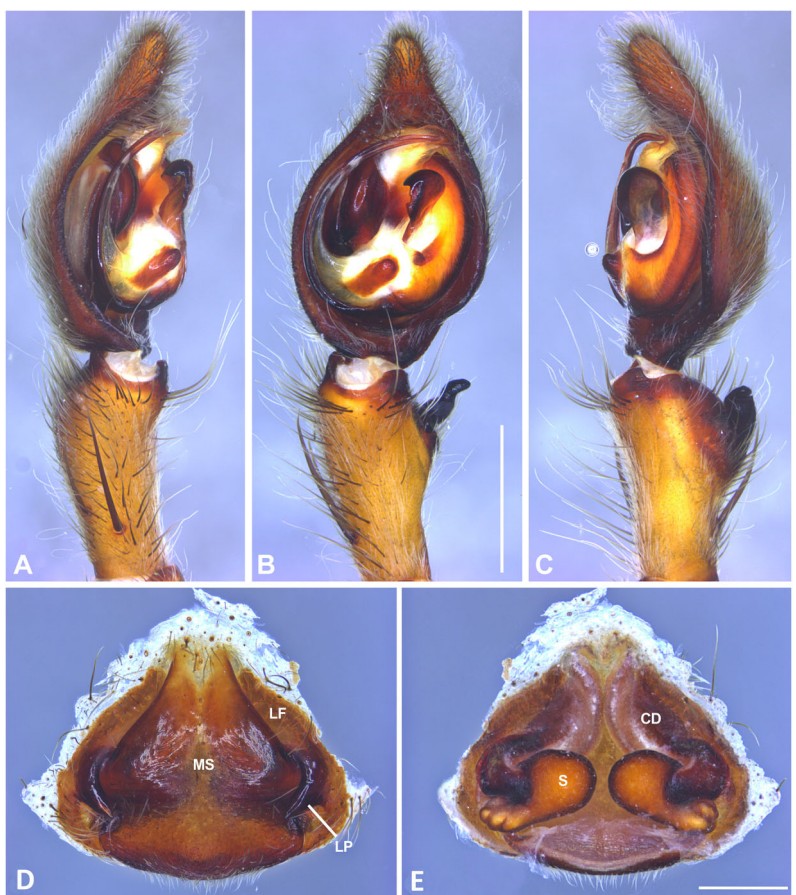

**Figure 10** *Kiekie curvipes* from Los Tuxtlas Biological Station (Costa Rica), left male palp. (A) Ventral view; (B) prolateral view; (C) retrolateral view, arrow indicates small peak of the median apophysis; scale bar = 2.00 mm; (D) epigynum, ventral view; (E) vulva, dorsal view. CD, copulatory duct; FD, fertilization duct; LF, epigynal lateral field; MS, epigynal median sector; S, spermatheca. Scale bar = 1.00 mm.                                               

measured by the area under the curve (AUC) metric, a method that evaluates predictive accuracy against a non-informative model on a scale that is independent of any specific decision threshold. The average AUC score across all replicates was calculated, and confidence intervals were established to discern the models' significance against what would be expected by chance (with AUC > 0.5 as the benchmark). For constructing binary distribution maps, we translated habitat suitability scores into binary presence-absence data, employing the 5th percentile as the cutoff point (*Liu et al., 2005*; *Liu, White & Newell, 2013*). Furthermore, we omitted regions that exhibited a high likelihood of species presence but were geographically isolated from documented specimen locations to refine the predictive maps (*Helgen et al., 2013*). For species of *Kiekie* with less than five records, we generated minimum convex polygon corrected by elevation range. For species with less than three records, we generated buffers of 10 km$^2$ for each record.

Species richness and endemism maps were generated in Biodiverse package (*Laffan, Lubarsky & Rosauer, 2010*) based on the distribution ranges estimated for each *Kiekie* species described above and using 0.05° of spatial resolution. Endemism maps were

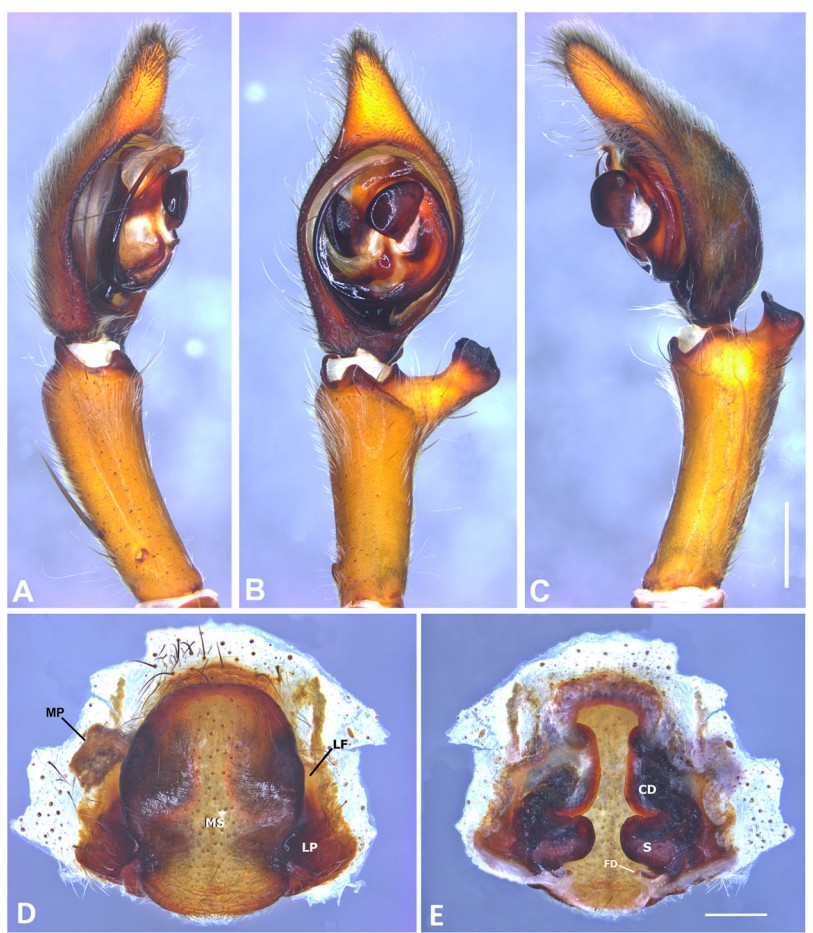

**Figure 11 *Kiekie garifuna*, left male palp.** (A) Ventral view; (B) prolateral view; (C) retrolateral view, arrow indicates small peak of the median apophysis; scale bar = 2.00 mm; (D) epigynum, ventral view; (E) vulva, dorsal view. CD, copulatory duct; FD, fertilization duct; LF, epigynal lateral field; MS, epigynal median sector; MT, mating plug; S, spermatheca. Scale bar = 1.00 mm.

computed using weighted endemism. In this metric, species richness is weighted by the inverse of the distribution range size of each species, so that pools of species that occur over smaller ranges are given higher scores. In addition, weighted endemism corrects for unweighted species richness, to distinguish per-species endemism from richness-based endemism (*Crisp et al., 2001*).

## HISTORICAL BIOGEOGRAPHY ANALYSIS

Based on the geographical distribution patterns of the species of *Kiekie*, we conducted the analysis using the following biogeographical units: Trans-Andean, Lower Central America, Talamanca Cordillera and Tropical North America. These regions are divided by well-known topographical and climatic barriers for other taxa in Central America (*Bagley & Johnson, 2017*; *Mendoza et al., 2019*), and additionally supported by the distribution patterns of *Kiekie* species. We used the maximum clade credibility dated tree resulting from the BEAST analysis considering only *Kiekie* and its sister group *Eldivo* gen. nov. for

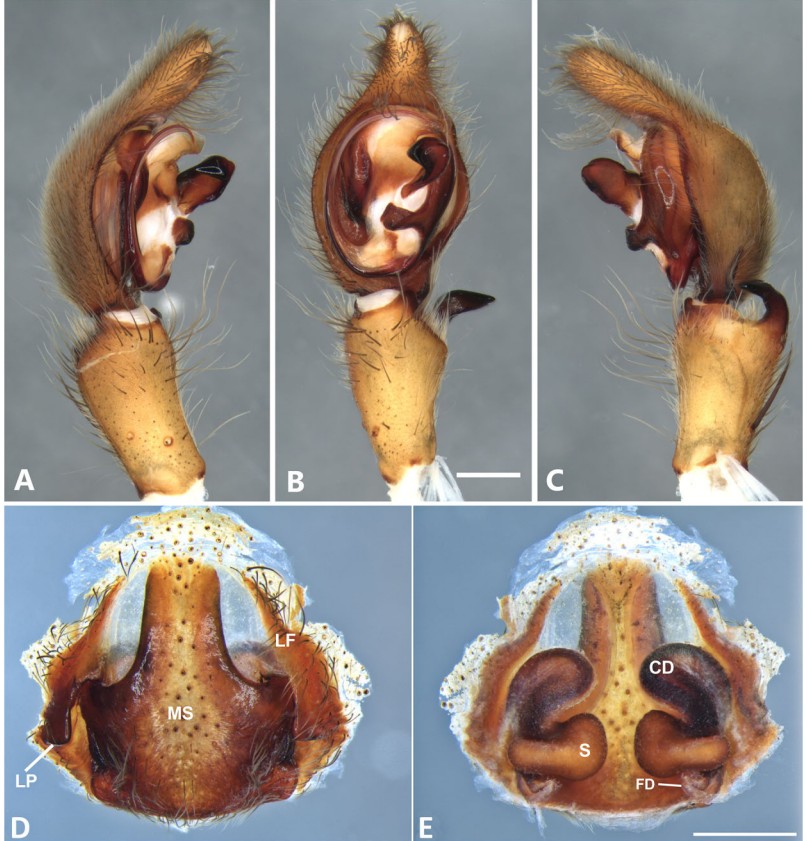

**Figure 12** *Kiekie griswoldi* **northern population (Villablanca Hotel), left male palp.** (A) Ventral view; (B) prolateral view; (C) retrolateral view; scale bar = 1.00 mm; (D) epigynum, ventral view; (E) vulva, dorsal view. CD, copulatory duct; FD, fertilization duct; LF, epigynal lateral field; MS, epigynal median sector; S, spermatheca. Scale bar = 0.50 mm; *Kiekie griswoldi* southern population (*Las Cruces Biological Station*), female genitalia.               

biogeographic analyses using the RASP package (*Yu et al., 2015*). We tested six biogeographic models using the Akaike information criterion (AICc): DIVA (*Ronquist, 1997*), DEC (*Ree & Smith, 2008*) and BayAREA (*Landis et al., 2013*). Each model was run with and without the founder-speciation event (j) (*Matzke, 2014*). The statistical comparisons of biogeographic models with and without the inclusion of the J parameter has been previously criticized upon the argument that the J parameter artificially inflates the contribution of cladogenetic events to the likelihood, and leads to underestimates of anagenetic, time-dependent ranges evolution (*Ree & Sanmartín, 2018*). However, models without the J parameters are usually inadequate in the common situations when most species have narrowed geographical ranges (single areas) (*Matzke, 2022*). Additionally, statistical comparison between models with and without the J parameter are identical to comparison of two ClaSSE submodels (*Matzke, 2022*). Therefore, we considered valid the used and statistical comparison between biogeographic models with and without the inclusion of the J parameter.

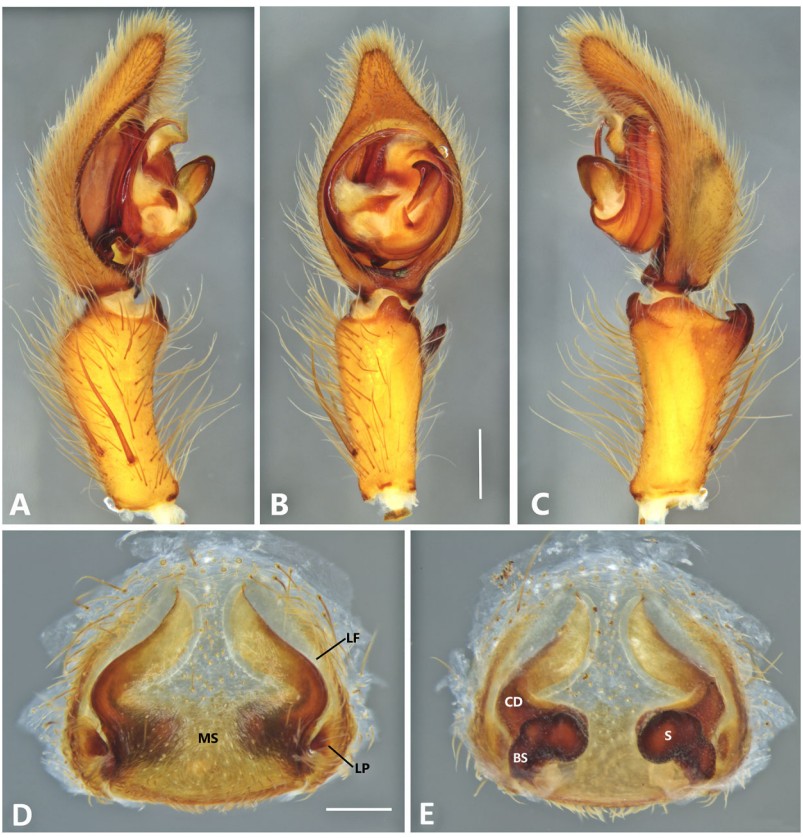

**Figure 13** *Kiekie lamuerte* **sp. nov, left male palp.** (A) Ventral view; (B) prolateral view; (C) retrolateral view; scale bar = 1.00 mm; (D) epigynum, ventral view; (E) vulva, dorsal view. BS, base of the spermatheca; CD, copulatory duct; FD, fertilization duct; LF, epigynal lateral field; MS, epigynal median sector; S, spermatheca. Scale bar = 0.50 mm. 

## Nomenclatural acts

According to the International Commission on Zoological Nomenclature (ICZN), the electronic version of this article in portable document format (PDF) will represent a published work. The new species names contained in the electronic version are effectively published under that Code from the electronic edition alone. This article and the nomenclatural acts it contains have been registered in ZooBank, the online registration system for the ICZN. The ZooBank LSIDs (Life Science Identifiers) can be resolved and the associated information viewed through any standard web browser by appending the LSID to the prefix http://zoobank.org/. The LSID for this publication is: urn:lsid:zoobank.org:pub:A693D4C7-C4DD-4F69-9D2A-DDD62527B4CF.

# RESULTS

## Phylogenetics

The likelihood analysis indicated that *Kiekie* is monophyletic with high support (Fig. 1) and placed within a clade of North and Central American ctenids of the genera *Leptoctenus* and *Ctenus*. The parsimony analysis also showed *Kiekie* as monophyletic but with low support (Fig. S1). The sister group of *Kiekie* is an undescribed species from Mexico which

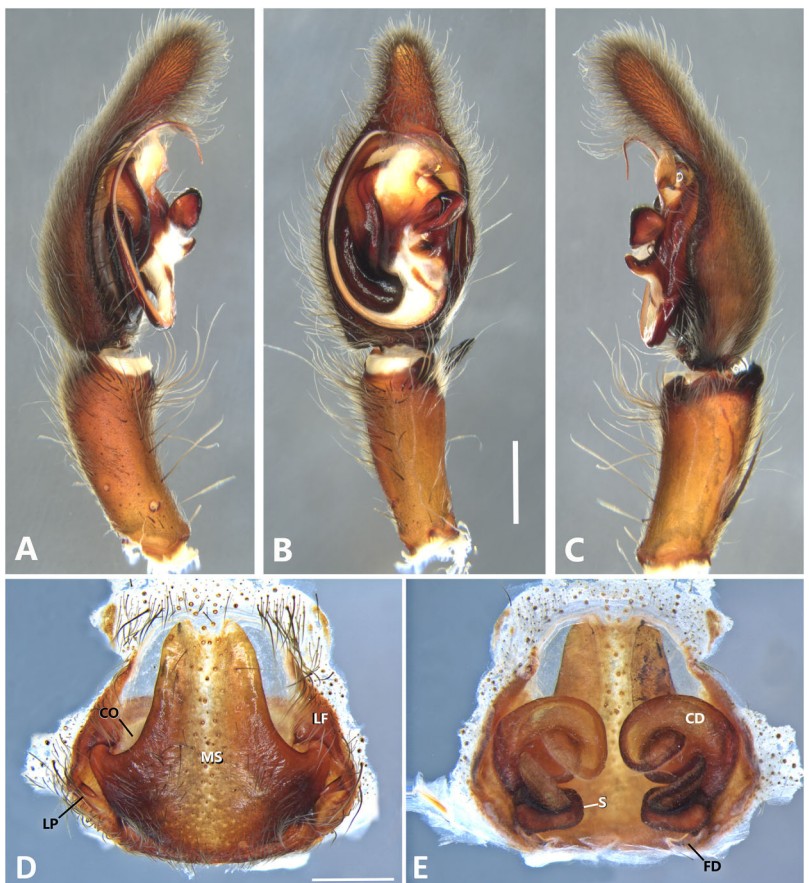

**Figure 14** *Kiekie laselva* **sp. nov, left male palp.** (A) Ventral view; (B) prolateral view; (C) retrolateral view; scale bar = 2.00 mm; (D) epigynum, ventral view; (E) vulva, dorsal view. CD, copulatory duct; CO, copulatory opening; FD, fertilization duct; LF, epigynal lateral field; MS, epigynal median sector; S, spermatheca. Scale bars = 1.00 mm.

lacks *Kiekie*'s morphological synapomorphies and diagnostic characters, and therefore is proposed here as a new genus (*Eldivo* gen. nov.). This sister group relationship is highly supported in both the likelihood and parsimony analyses. Both analyses also show several well-supported intrageneric groups in *Kiekie*. *Kiekie garifuna* is placed sister to the remaining species in the genus and two well-supported clades are identified. The first clade includes the species *Kiekie barrocolorado*, *K. verbena*, *K. montanense* and *K. curvipes* (Keyserling, 1881). Within this clade the sister relationship between *K. curvipes* and *K. montanense* is highly supported. The second clade includes the species *K. sarapiqui Polotow & Brescovit, 2018*, *K. valerioi* sp.nov, *K. laselva* sp.nov, *K. panamensis*, and *K. griswoldi*.

## Biogeography

The species richness map indicated that *Kiekie* reach its highest diversity in montane ecosystems of Costa Rica, followed by the lowlands of the Pacific area of Costa Rica (Fig. 2A). In addition, most of the endemism of this genus is concentrated in the montane ecosystems of Costa Rica (Fig. 2B). Model testing provided stronger support for models

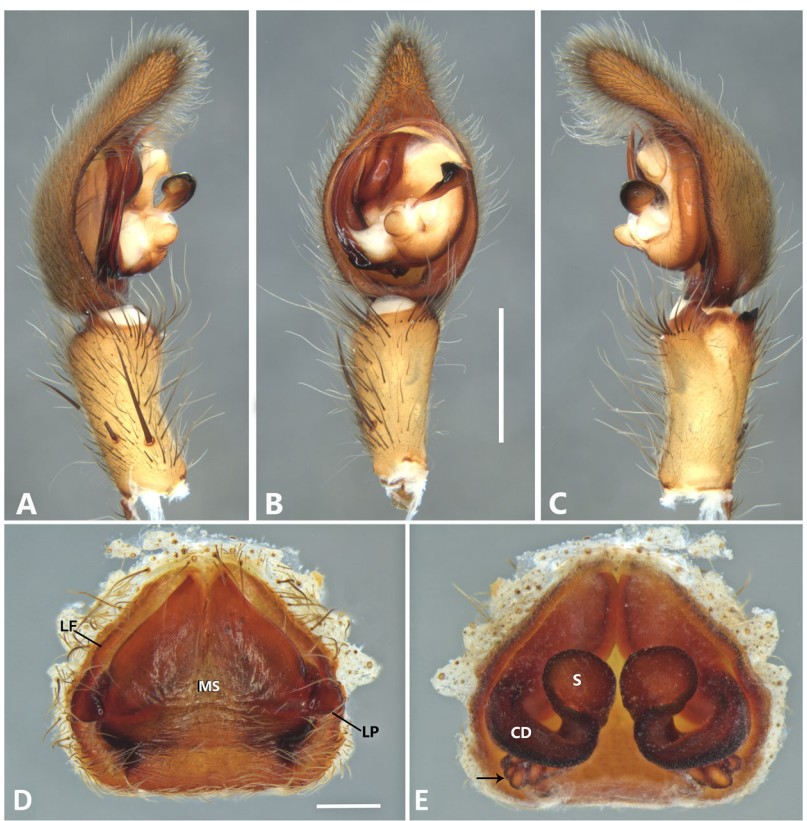

**Figure 15 _Kiekie montanensis_ sp. nov, left male palp.** (A) Ventral view; (B) prolateral view; (C) retrolateral view, arrow indicates small peak of the median apohysis; scale bar = 2.00 mm; (D) Epigynum, ventral view; (E) Vulva, dorsal view, black arrow indicates numerous granules below of the copulatory ducts. CD, copulatory duct; FD, fertilization duct; LF, epigynal lateral field; MS, epigynal median sector; S, spermatheca. Scale bar = 0.50 mm.

with the jump dispersal parameter over models without it. Models with the j parameter gave similar results in the ancestral area reconstruction for _Kiekie_, and they could not be statistically distinguished from each other (AIC weights: DEC + J = 0.38, DIVA + J = 0.35, BAYESAREA + j = 0.28). These analyses support an origin of _Kiekie_ during the Late Miocene in the Tropical North America region and subsequently dispersed to Central America, where its highest diversity arises (Figs. 3A and 3B, event 1). The following description of the historical biogeography of _Kiekie_ is based on the results of DEC + J model. One event of dispersal (event 2) from Talamanca Cordillera to Lower Central America originated two species: _Kiekie tirimbina_ sp.nov and _K. barrocolorado_. Then, there was an event of dispersal (event 3) from Lower Central America back to the cordillera Talamanca originating the species _K. verbena_ and _K. montanense_. The widespread species _K. curvipes_ originated from the latter lineage of montane species, dispersing into the lowlands of Central and Tropical North America (event 4). The clade comprising the montane species _K. sinuatipes_ and _K. barrantesi_ sp.nov likely originated from the lowlands of Central America (event 5) but there is a high uncertainty in the area reconstruction of the node preceding this lineage. Finally, there were two dispersal events (event 6) from the lowlands of Central America to Talamanca Cordillera and the Trans-Andean region,

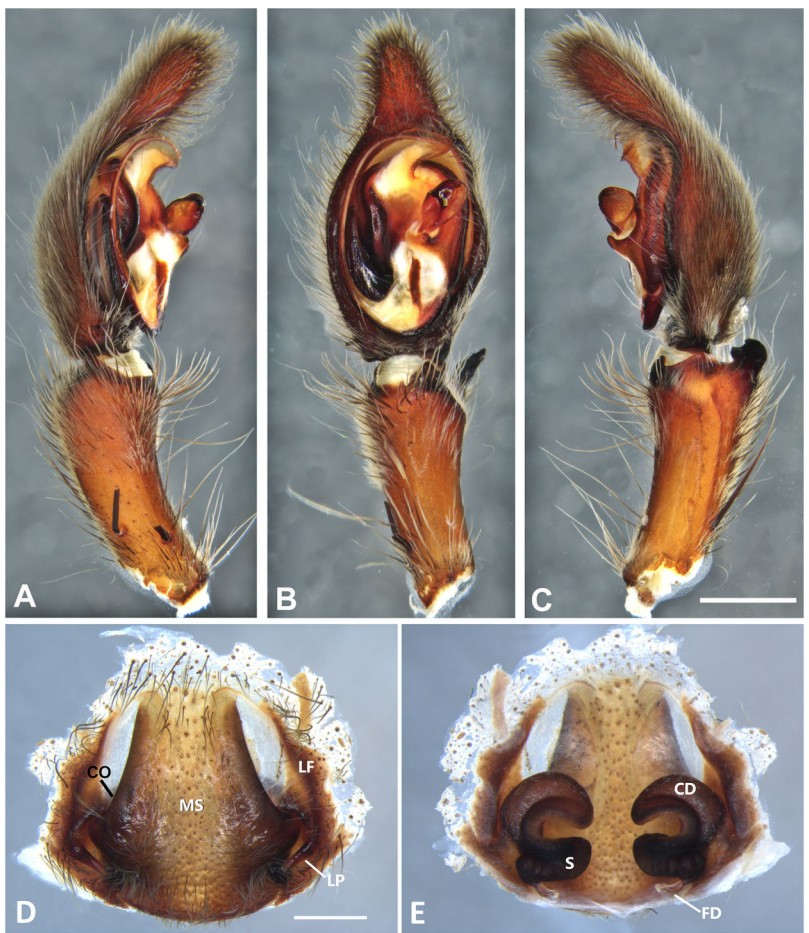

**Figure 16** *Kiekie panamensis*, **left male palp.** (A) Ventral view; (B) prolateral view; (C) retrolateral view, arrow indicates small peak of the median apophysis; scale bar = 2.00 mm; (D) epigynum, ventral view; (E) vulva, dorsal view. CD, copulatory duct; CO, copulatory opening; FD, fertilization duct; LF, epigynal lateral field; MS, epigynal median sector; S, spermatheca. Scale bar = 1.00 mm.

originating the montane clade *K. lascruces* and *K. griswoldi*, and the Central and South America species *K. panamensis*, respectively.

## Taxonomy

**Family Ctenidae Keyserling, 1877**

**Subfamily Cteninae Keyserling, 1877**

**Genus *Kiekie* Polotow & Brescovit, 2018**
Type **species**: *Ctenus sinuatipes* F.O. Pickard-Cambridge, 1897.

**Composition**: 18 species: *Kiekie sinuatipes* (F.O. Pickard-Cambridge, 1897), *K. curvipes* (Keyserling, 1881), *K. garifuna* Polotow & Brescovit, 2018, *K. verbena* Polotow & Brescovit, 2018, *K. sarapiqui* Polotow & Brescovit, 2018, *K. griswoldi* Polotow & Brescovit, 2018, *K. barrocolorado* Polotow & Brescovit, 2018, *K. panamensis* Polotow & Brescovit, 2018,

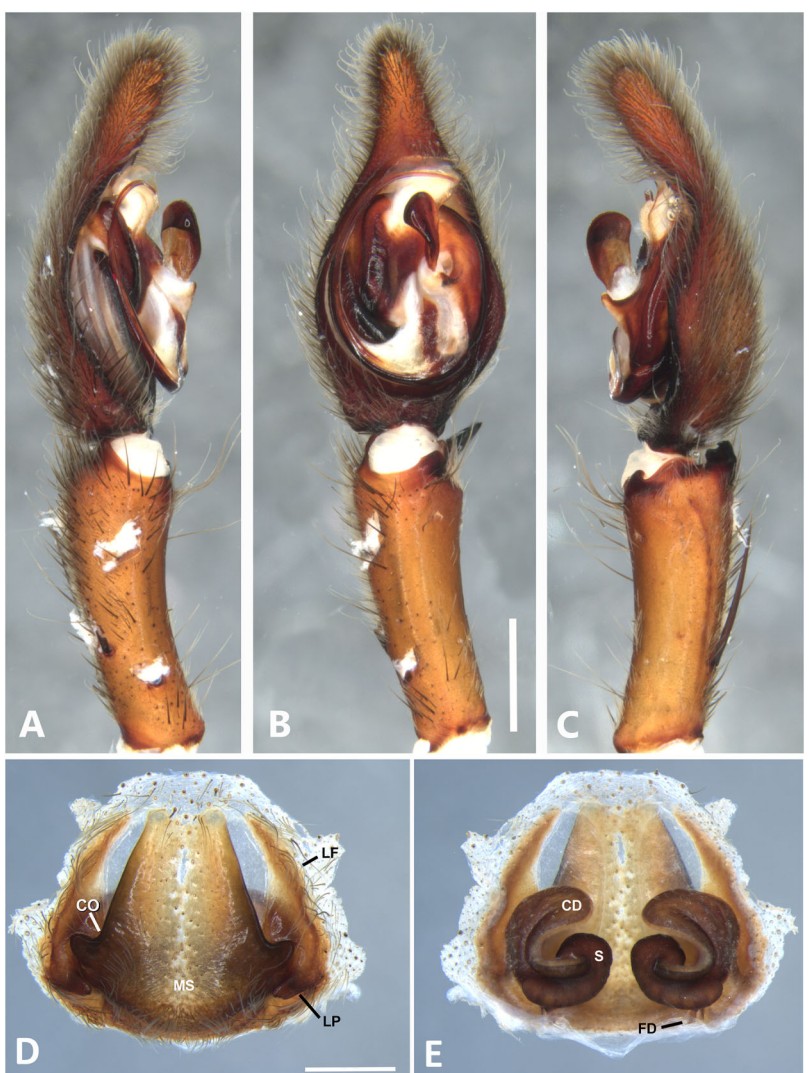

**Figure 17 Kiekie sarapiqui, left male palp.** (A) Ventral view; (B) prolateral view; (C) retrolateral view, arrow indicates small peak of the median apophysis; scale bar = 2.00 mm; (D) epigynum, ventral view; (E) vulva, dorsal view. CD, copulatory duct; CO, Copulatory opening; FD, fertilization duct; LF, epigynal lateral field; MS, epigynal median sector; S, spermatheca. Scale bar = 1.00 mm.

*K. montanense* Polotow & Brescovit, 2018, and *K. antioquia* Polotow & Brescovit, 2018, *K. almae* Omelko, 2023, *K. dietrichi* Omelko, 2023 *K. laselva* sp. nov., *K. valerioi* sp. nov., *K. barrentesi* sp. nov., *K. lamuerte* sp. nov., *K. bernali* sp. nov., *K. tirimbina* sp. nov.

**Diagnosis.** Males of *Kiekie* can be distinguished from the remaining Ctenidae by three synapomorphies: an elongated embolus with a laminar process (Figs. 23A–23F, 24A–24F), conductor resembling an open fan (Figs. 23A–23F, 25A–25F), and locking lobes located at posterior and sometimes in retro-posterior side (Figs. 25B, 26C), instead of posterior prolateral as most ctenines. Some females of *Kiekie* (*K. sinuatipes*, *K. barrantesi* sp.nov, *K. sarapiqui*, *K. laselva* sp.nov, *K. valerioi* sp.nov, *K. panamensis*, *K. griswoldi* and *K. lascruces*) differ from the remaining Ctenidae by elongate curved copulatory ducts that

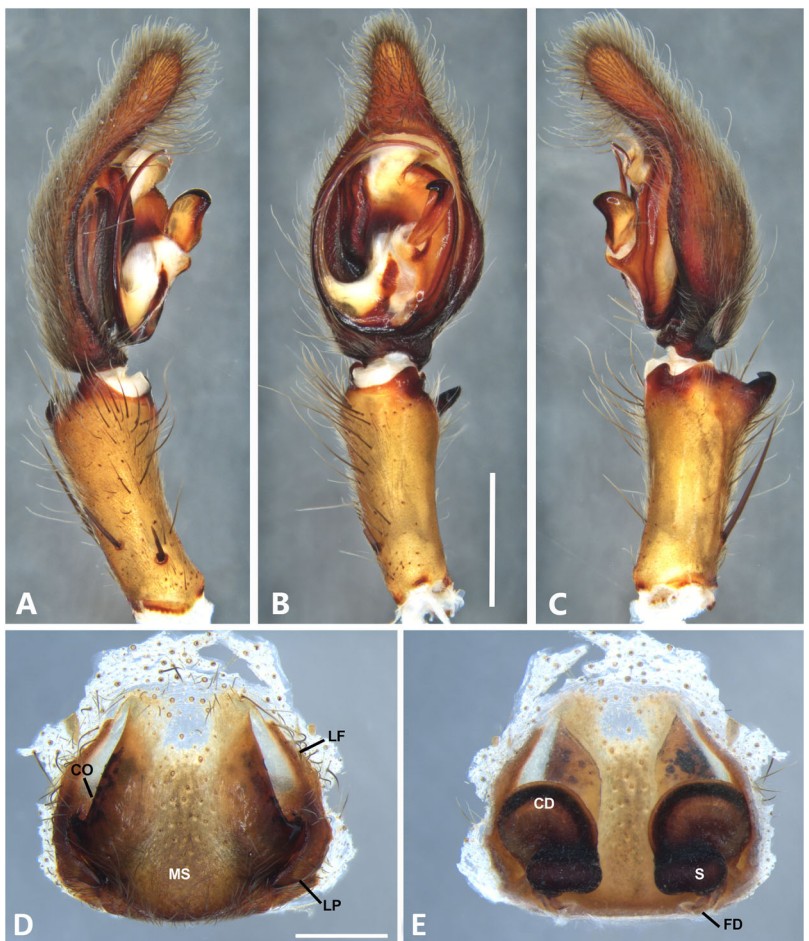

**Figure 18** *Kiekie sinuatipes*, **left male palp.** (A) Ventral view; (B) prolateral view; (C) retrolateral view, arrow indicates small peak of the median apophysis; scale bar = 2.00 mm; (D) epigynum, ventral view; (E) vulva, dorsal view. CD, copulatory duct; CO, copulatory opening; FD, fertilization duct; LF, epigynal lateral field; MS, epigynal median sector; S, spermatheca. Scale bar = 1.00 mm.

projects in the anterior area of the epigynum (Figs. 6E, 16E and 20E) as opposed to short and straight, and large copulatory openings instead of small. However, females of the remaining species cannot be accurately distinguished from *Eldivo* or other Mesoamerican ctenids.

**Natural History.** As most species of *Ctenus* and *Spinoctenus* (*Polotow & Brescovit, 2014*; *Hazzi et al., 2018*), males of *Kiekie* present a slightly to strongly curved metatarsus IV with various small macrosetae (Fig. 5B). In *Kiekie*, this modified male metatarsus is used during courtship (W. Ebehard, G. Barrantes & N. Conejo, 2019, personal communication). Moreover, in several species of *Kiekie* (*e.g.*, *K. panamensis Polotow & Brescovit, 2018*, *K. laselva* sp.nov, *K. sinuatipes* (F.O. Pickard-Cambridge, 1897), and *K. valerioi* sp. nov), males and females have strikingly different coloration patterns. While females are mostly black with few gold or orange lines on the abdomen, males are pale brown with dark brown alternating reticular bands (Figs. 5A, 5C, 5E and 5F).

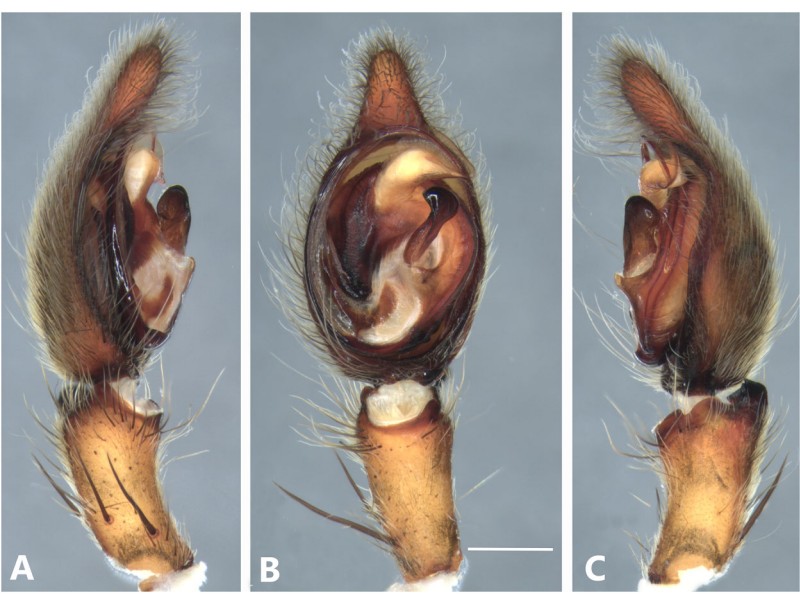

**Figure 19** *Kiekie tirimbina* **sp. nov, left male palp.** (A) Ventral view; (B) prolateral view; (C) retrolateral view; scale bar = 1.00 mm.

*Kiekie* species live in a variety of ecosystems and across a wide elevational range, from sea level to 2,500 m. As other ctenines, *Kiekie* species are nocturnal forest dwellers, living on ground covered with leaf litter and on some cases on vegetation in the understory. During the day, these spiders can be found hiding under leaf litter or decaying fallen logs. *Kiekie sarapiqui*, *K. valerioi* sp.nov, *K. curvipes*, *K. garifuna*, *K. tirimbina* sp.nov, *K. laselva* sp.nov and *K. panamensis* are found in lowland tropical rain forests (>700 m). However, *K. curvipes* and *K. panamensis* are widespread species that can be found in less humid ecosystems and higher elevations (~1,500 m). In Costa Rica, there are high levels of species sympatry in *Kiekie*. The most striking case occurs in the area of the Tirimbina Biological Station, where five species can be found in sympatry: *K. valeoroi*, *K. curvipes*, *K. sarapiqui*, *K. laselva* sp.nov and *K. tirimbina* sp.nov, the latter being rarely observed. *Kiekie valerioi* sp.nov and *K. sarapiqui* are more abundant in the preserved forest and *K. curvipes* is abundant in both preserved and disturbed environments. These species live also in sympatry with other ctenids, such as *Phoneutria depilata* (Strand, 1909) and the arboreal species *Phymatoctenus cf. sassii* Reimoser, 1939. In montane ecosystems of the Talamanca cordillera, there are eight species restricted to montane ecosystems: *K. bernali* sp.nov, *K. griswoldi*, *K. las cruces*, *K. lamuerte* sp.nov, *K. sinuatipes*, *K. barrantesi* sp.nov, *K. verbena*, and *K. montanense*. In the north of Costa Rica, it is possible to find in some localities *K. griswoldi*, *K. sinuatipes*, *K. verbena* and *K. barrantesi* sp.nov in sympatry. In the campus of Universidad of Costa Rica, *K. barrantesi* sp.nov and *K. verbena* are easily found in very disturbed environments. In the southern part of the cordillera, *K. lascruces* inhabits in sympatry with *K. montanense*. *Hazzi et al. (2020)* studied the effects of forest succession and microenvironmental variables on the abundance of ctenid spiders in Las Cruces Biological Station, a tropical montane rainforest. This study found four ctenids living in sympatry: *Spinoctenus escalerete Hazzi et al., 2018*, *K. griswoldi*, and three species

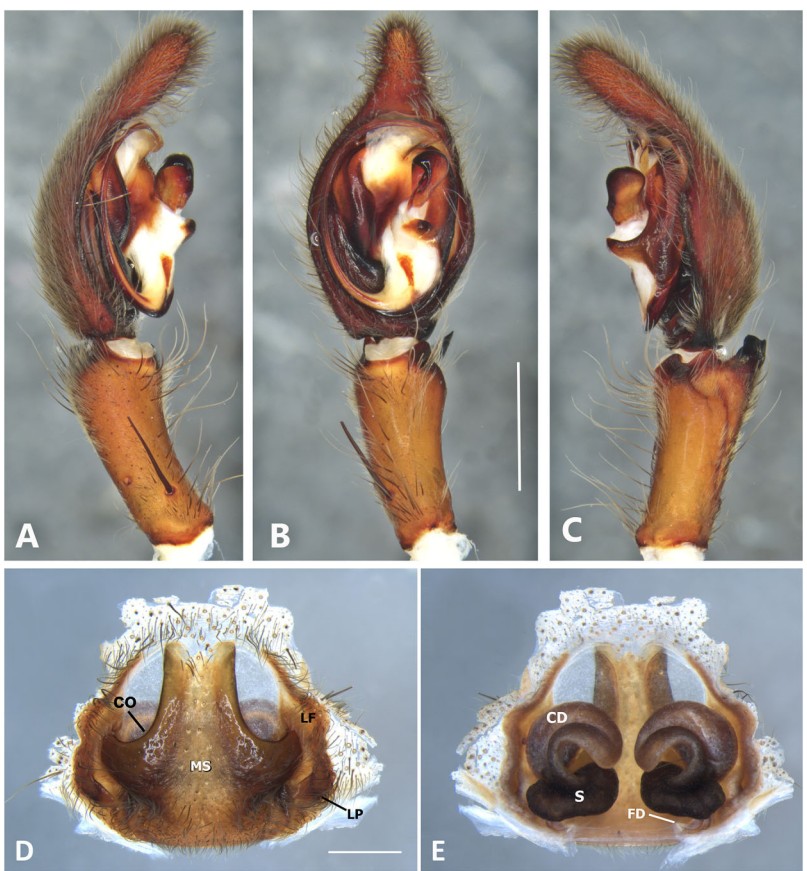

**Figure 20** *Kiekie valerioi* **sp. nov, left male palp.** (A) Ventral view; (B) prolateral view; (C) retrolateral view, arrow indicates small peak of the median apophysis; scale bar = 2.00 mm; (A) epigynum, ventral view; (B) vulva, dorsal view. CD, copulatory duct; CO, copulatory opening; FD, fertilization duct; LF, epigynal lateral field; MS, epigynal median sector; S, spermatheca. Scale bar = 1.00 mm.

with much lower abundances. The larger species, *K. griswoldi* was significantly more abundant in primary than in secondary forest. Conversely, *S. escalerete* was more abundant in the secondary forest. In addition, this study found that while abundance of *K. lascruces* was positively related with leaf litter depth, the abundance of *S. escalerete* was negatively related to litter depth. This study also reported an event of intraguild predation where an adult female of *K. griswoldi* was eating an adult male of *S. escalerete* in the primary forest. In Cali, Colombia, females of *K. panamensis* were observed with egg sacs in both captivity and in the field. The egg sacs were white with a flat face attached to the substrate (*i.e.*, leaf litter or the wall of the terrarium) and a convex face. In captivity, we observed that females protected the egg sacs by standing over them. Spiderlings emerged between 29–31 days (*n* = 2). After hatching, spiderlings emerged and built an irregular web where they remained until their second molt. These small observations about the reproductive behavior of *K. panamensis* are consistent with those reported in *Phoneutria depilata* (*Hazzi, 2014*).

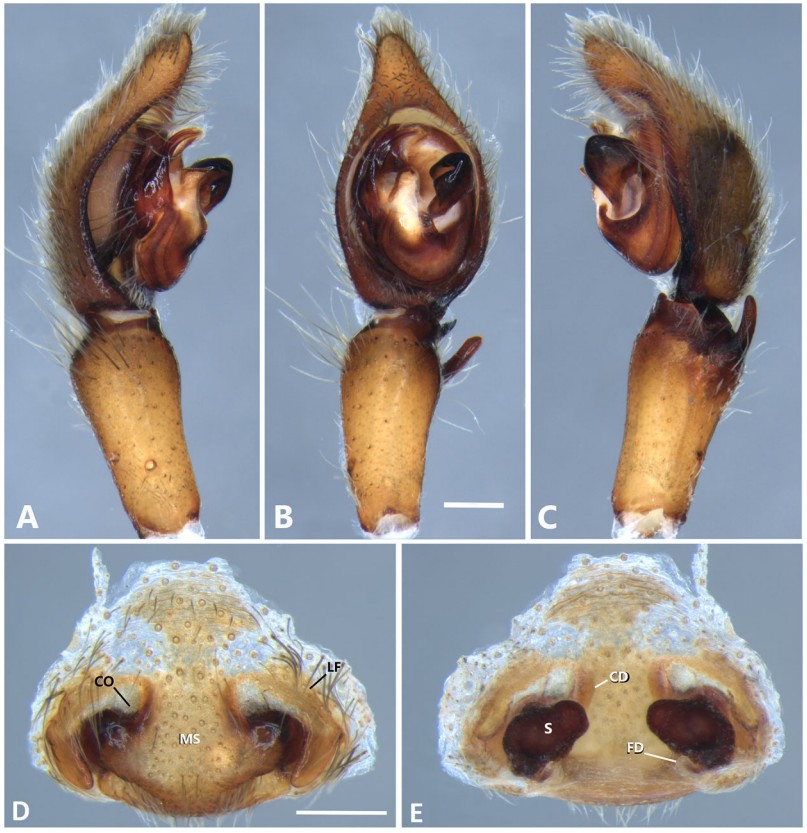

**Figure 21** *Kiekie verbena* **sp. nov, left male palp.** (A) Ventral view; (B) prolateral view; (C) retrolateral view, arrow indicates small peak of the median apophysis; scale bar = 0.5 mm; (D) epigynum, ventral view; (E) vulva, dorsal view. CD, copulatory duct; CO, copulatory opening; FD, fertilization duct; LF, epigynal lateral field; MS, epigynal median sector; S, spermatheca. Scale bar = 0.50 mm.

### *Kiekie antioquia* Polotow & Brescovit, 2018

*Kiekie antioquia* Polotow & Brescovit, 2018: 370, f. 14A-B (Df). Female holotype from Colombia, Antioquia Department, Remedios (7.02, −74.69), deposited in MCZ30617 (not examined).

**Diagnosis.** Females of *Kiekie antioquia* (fig. 14A–B, *Polotow & Brescovit, 2018*) resemble those of *K. sinuatipes* (fig. 5A–B, *Polotow & Brescovit, 2018*) by the large median field of the epigynum, but it can be distinguished by the narrow median field and elongated and narrow lateral projections. Internally, it can be distinguished by smaller spermathecae and sinuous copulatory ducts (*Polotow & Brescovit, 2018*).

**Female.** Described by *Polotow & Brescovit (2018)*.
Distribution. Department of Antioquia, Colombia.

### *Kiekie almae* Omelko, 2023

*Kiekie almae* Omelko, 2023: 276, f. 1–5, 11–14, 19–22, 27–30, 36–37. Male holotype and three female paratypes from Panama, Chiriquí Province, Totumas Mt., 8.8842N 82.6780W, deposited in ZMMU, Moscow, (not examined).

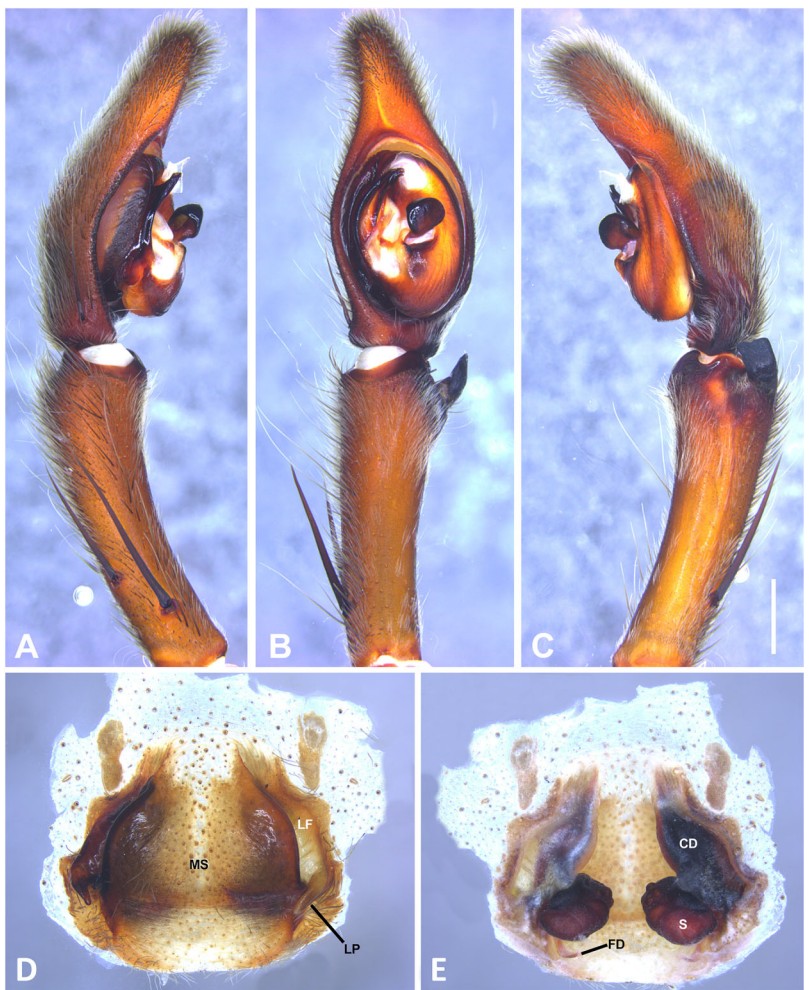

**Figure 22 *Eldivo tuxtlas* sp. nov, left male palp.** (A) Ventral view; (B) prolateral view; (C) retrolateral view, arrow indicates small peak of the median apohysis; scale bar = 1.00 mm; (D) Epigynum, ventral view; (E) Vulva, dorsal view. CD, copulatory duct; CO, copulatory opening; FD, fertilization duct; LF, epigynal lateral field; MS, epigynal median sector; S, spermatheca. Scale bar = 0.5 mm.

**Diagnosis.** Males of *K. almae* resemble those of *K. lamuerte* sp. nov, in the general conformation of the palp (Fig. 22; *Omelko, 2023*: fig. 12) but differ by the medial position of the RTA (*Omelko, 2023*, fig: 12), in contrast with the more apical RTA in *K. lamuerte* sp.nov (Fig. 13). Females of *K. almae* resemble those of *K. garifuna* in the wide anterior area of the epigynal median sector (Fig. 13D; *Omelko, 2023*: figs. 27 and 28), but differ by being squarer and flatter, in contrast with the ovoid and elevated anterior area in *K. garifuna* (modified from *Omelko, 2023*).

**Male.** Described by *Omelko (2023)*.

**Distribution.** Known only from the type locality in the Chiriquí Province of Panama.

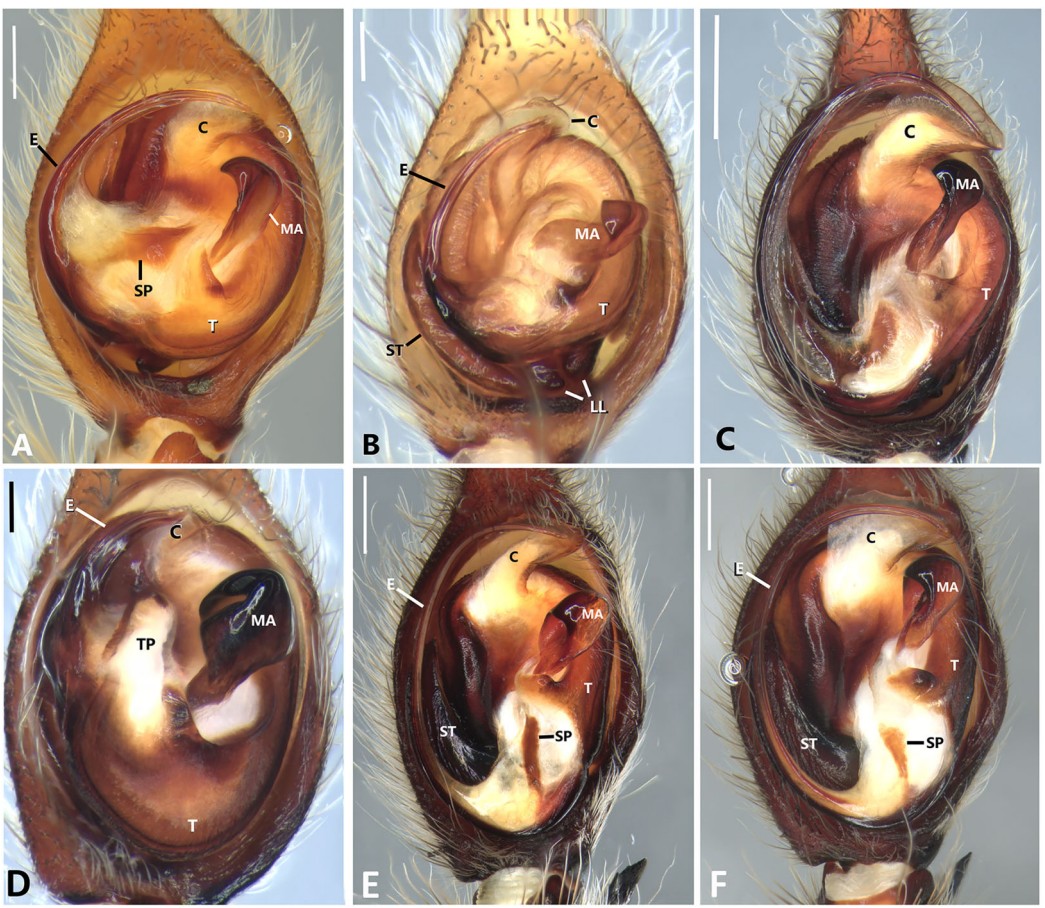

**Figure 23 Left male palp of *Kiekie* spp.** Ventral view of left male palp of *Kiekie* spp. (A) *Kiekie lamuerte* sp.nov; (B) *K. bernali* sp.nov; (C) *K. tirimbina* sp.nov; (D) *K. verbena*; (E) *K. panamensis*; (F) *K. valerioi* sp.nov. Abbreviations: C, conductor; E, embolus; LL, locking lobes, median sector; MA, median apophysis; SP, sclerotized process at the base of the embolus; ST, subtegulum; t, tegulum; TP, tegular process. Scales bars: 0.50 mm (A), 0.20 mm (B), 1.00 mm (C), 0.20 mm (D), 1.00 mm (E), 1.00 mm (F).

### *Kiekie barrantesi* sp. nov.
Figures 6, 24B and 26B.

## Type material
*Holotype*. COSTA RICA: Male from San José province, San Pedro, campus of Universidad of Costa Rica (9.9376N, 84.0507W, 1,200 m), 10.VI-2018, N. Hazzi (MCZ IZ 167568).
*Paratypes*. Three males and three females, same data as holotype; two females from William and Mary Jane Eberhard house, San Antonio de Escazu, San José province (9.8975N, 84.1377W, 1,330 m), 20.VII.2018, N. Hazzi (MCZ IZ 167569).

*Other material examined*. COSTA RICA: San José province: One male and one female, San José, Sabanas Sur (9.9308N, 84.0933W), 12.V.1981, G. Biamonte (UCR); one female and one male, San Rafel de Moravia (9.9688N, 84.0489W), 00.V.1963, C. Valerio (UCR); Tibas, Parques del Norte (9.9666N, 84.0736W), C. Herrera (MNCR); Alajuela province: One Male, Palmares (10.0607N, 84.4377W), 20.IV.2013, S. Zamora; Cartago province: One

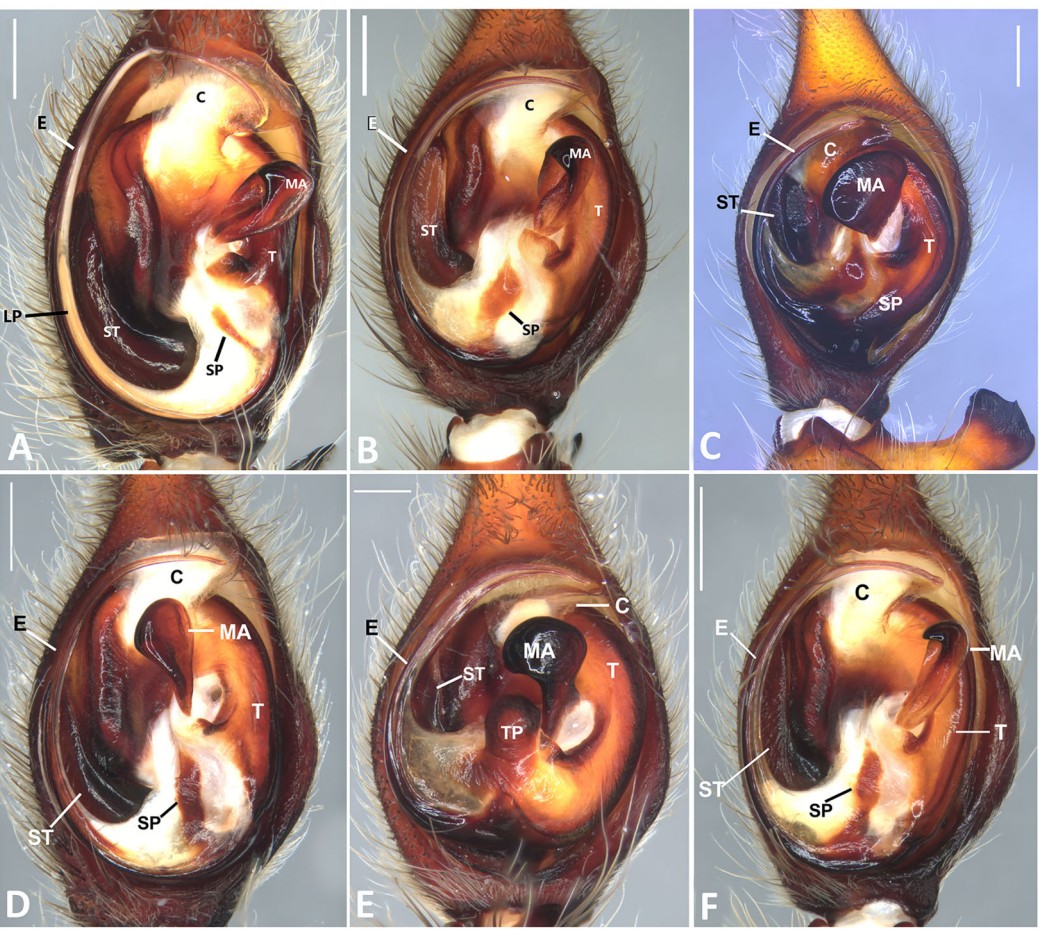

**Figure 24 Left male palp of *Kiekie* spp.** Ventral view of left male palp of *Kiekie* spp. (A) *K. laselva* sp. nov; (B) *K. barrantesi* sp.nov; (C) *K. garifuna*; (D) *K. sarapiqui*; (E) *K. curvipes*; (F) *K. sinautipes*. Abbreviations: C, conductor; E, embolus; LL, locking lobes, median sector; MA, median apophysis; SP, sclerotized process at the base of the embolus; ST, subtegulum; t, tegulum; TP, tegular process. Scales bars: 1.00 mm (A), 1.00 mm (B), 1.00 mm (C), 0.5 mm (D), 1.00 mm (E), 1.00 mm (F).

female, Jimenez, Pejibaye, Copal Biological Station (9.7834N, 83.7519W), 00.IV.2005 (MNCR); Heredia province: One female, Calle los Gemelos (9.9834N, 84.1136W), 27. IV.1990, J. Carvajal (MNCR); San Joaquín, Flores (10.0062N, 84.1537W), 14.IV.1997 (MNCR); two males, Santo Domingo, Santa Rosa (9.9736N, 84.0944W), 15.II.1995, C. Viquez (MNCR). PANAMA: Chiriquí province: One male and one female, Fortuna Reserve (8.711N, 82.171W, 1,800 m), 4.XII.2018, N. Hazzi (MCZ IZ 167570).

**Etymology.** This species is dedicated to Gilbert Barrantes, professor from Universidad of Costa Rica, who has made important contributions to the study of spider behavior and natural history.

**Diagnosis.** Males resemble those of *K. panamensis* and *K. valerioi* sp.nov by the general configuration of the palp, but can be distinguished from them by the orientation of the sclerotized process at the base of the embolus (Figs. 6B and 24B), circular shaped-bulb

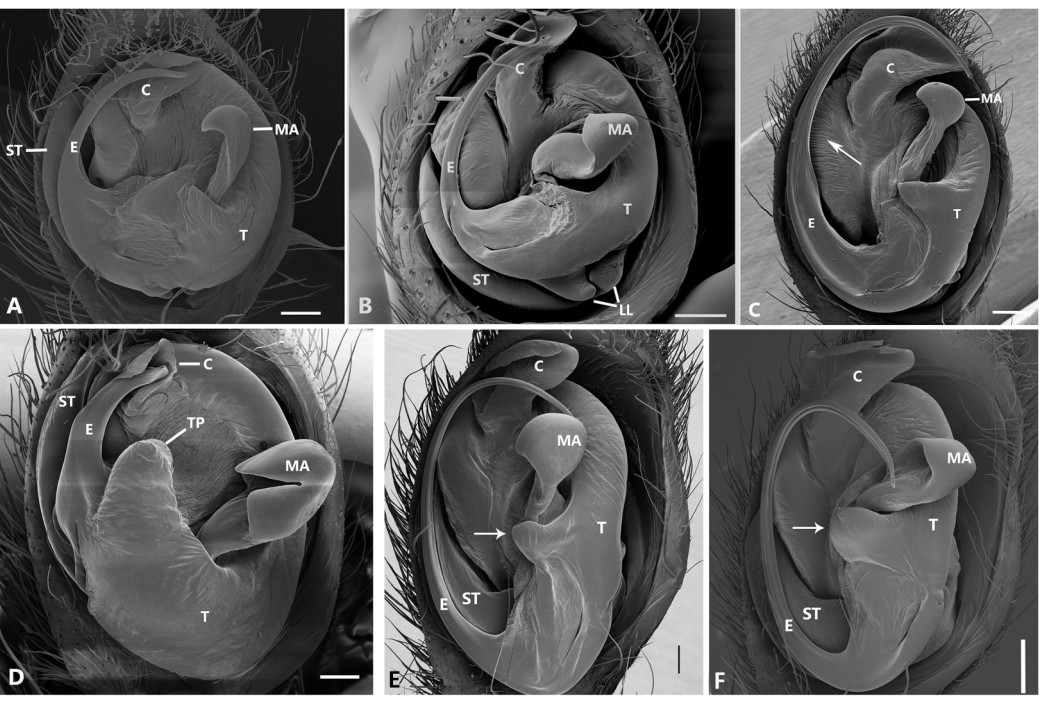

**Figure 25** *Ventral view of left male palp of Kiekie* **spp.** (A) *Kiekie lamuerte* sp.nov; (B) *K. bernali* sp.nov; (C) *K. tirimbina* sp.nov; (D) *K. verbena;* (E) *Kiekie panamensis;* (F) *K. valerioi* sp.nov. Abbreviations: C, conductor; E, embolus; LL, locking lobes, median sector; MA, median apophysis; ST, subtegulum; t, tegulum; TP, tegular process. Arrow in C indicates small grooves of the tegulum, and arrows in E and F indicate tegular projection at the base of the median apophysis. Scales bars: 0.20 mm (A), 0.10 mm (B), 0.20 mm (C), 0.10 mm (D), 0.20 mm (E), 0.40 mm (F).

(Figs. 6B and 26B) (the other two species have oval bulbs being longer than wide) and a small RTA (Figs. 6B, 6C and 24B). Females resemble those of *K. sinuatipes* by the conformation of the copulatory ducts (Figs. 6E and 18E) but can be distinguished from them by an internal projection of the spermatheca (Fig. 6E), and anterior area of the median sector narrow (Fig. 6D) instead of wide.

*Male* (MCZ IZ 167568 from Campus of Universidad of Costa Rica, San Pedro, San José province, Costa Rica). Total length 27.00. Carapace 16.62 long and 12.08 wide. AME 0.60 ALE 0.52, PME 0.71, PLE 0.76. Sternum 6.73 long and 5.88 wide, labium 1.92 long and 1.70 wide, endites 2.93 long and 1.88 wide. Leg measurements: I: femur 18.81, patella 6.85, tibia 22.4, metatarsus 17.40, tarsus 6.12, total; II: 19.44, 6.27, 20.21, 16.55, 6.50, total; III: 15.83, 6.03, 15.95, 16.17, 5.50, total; IV: 20.40, 6.10, 20.50, 22.90, tarsus (missing), total. Leg spination: leg I-II tibia v2-2-2-2-2, d0-0-1-0, p1-1-0, r1-1-0, metatarsus v2-2-2; leg III-tibia v2-2-2, d1-1-1, p1-0-1, r1-0-1, metatarsus v2-2-2, d0-1-0, p1-1-2, r1-1-2; leg IV-tibia v-2-2-2, d1-1-1, p1-01, r1-0-1, metatarsus modified. Palp: RTA spiniform in ventral view and with wide apex (Fig. 6C), embolus elongated and flagelliform, presence of sclerotized process at the base of the embolus (Figs. 6B and 24B), median apophysis with cup-shaped aperture visible ventrally (Figs. 24B and 26B); conductor with a narrow base and large apex, covering the tip of the embolus (Figs. 6B, 24B and 26B).

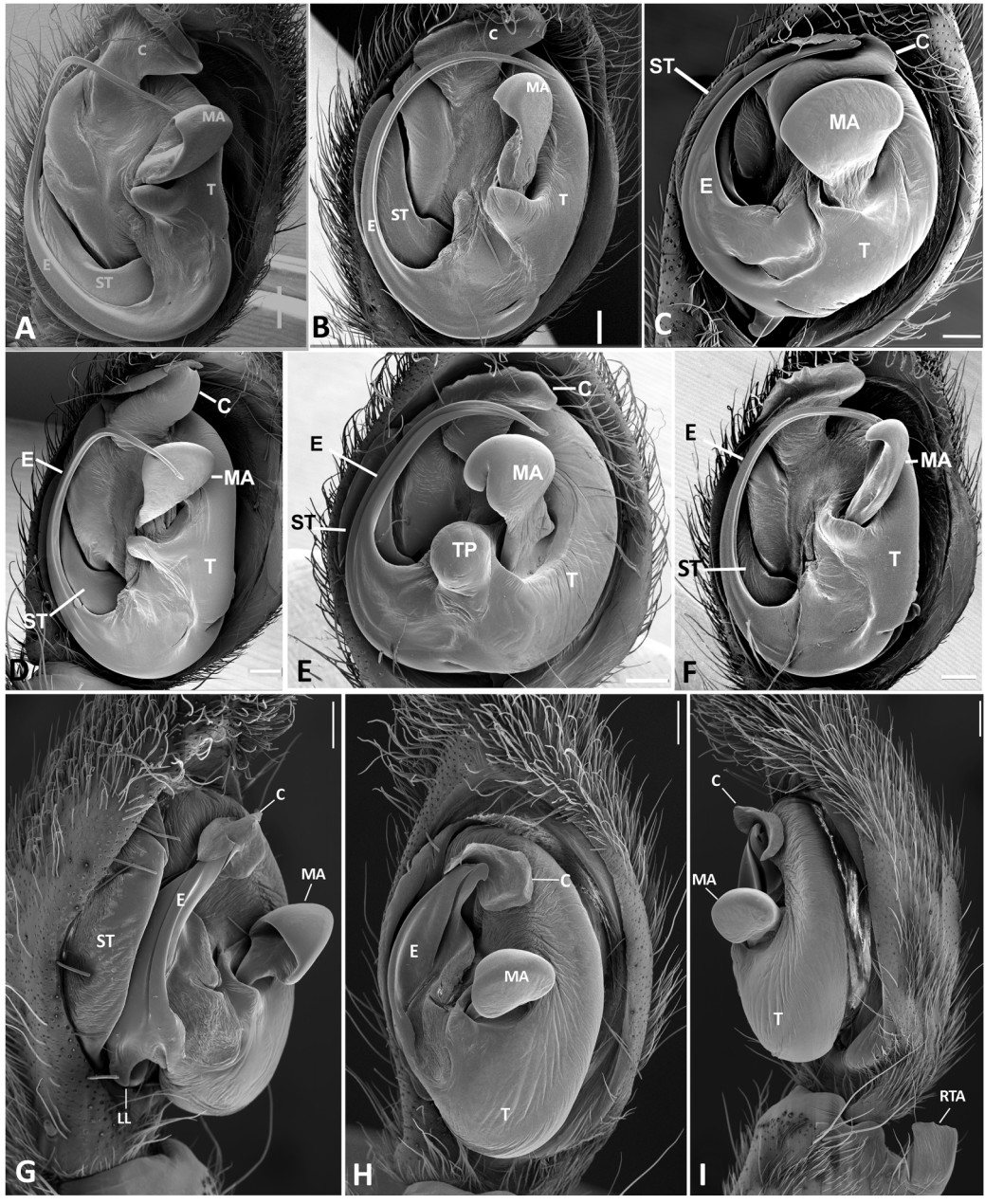

**Figure 26 Ventral view of left male palp of *Kiekie* spp.** (A) *K. laselva* sp.nov; (B) *K. barrantesi* sp.nov; (C) *K. garifuna*; (D) *K. sarapiqui*; (E) *K. curvipes*; (F) *K. sinautipes*; *Eldivo tuxtlas* sp. nov, left male palp. (G), ventral view; (H), prolateral view; (I), retrolateral view, arrow indicates small peak of the median apophysis White arrows in A and B, show the larger tegular process of *K. valerioi* sp.nov compared to *K. panamensis*. Abbreviations: C, conductor; E, embolus, median sector; MA, median apophysis; ST, subtegulum; t, tegulum. Scales bars: 0.20 mm (A), 0.40 mm (B and C), 0.20 mm (D), 1.00 mm. (G–I).

*Female* (MCZ IZ 167569-1, from Campus of Universidad of Costa Rica, San Pedro, San José province, Costa Rica). Total length 35.18. Carapace 16.22 long and 12.64 wide. AME 0.70, ALE 0.50, PME 0.80, PLE 0.80. Sternum 6.73 long and 6.06, labium 2.27 long and 2.13 wide, endites 4.61 long and 2.50 wide. Leg measurements: I: femur 15.16, patella 7.10. tibia

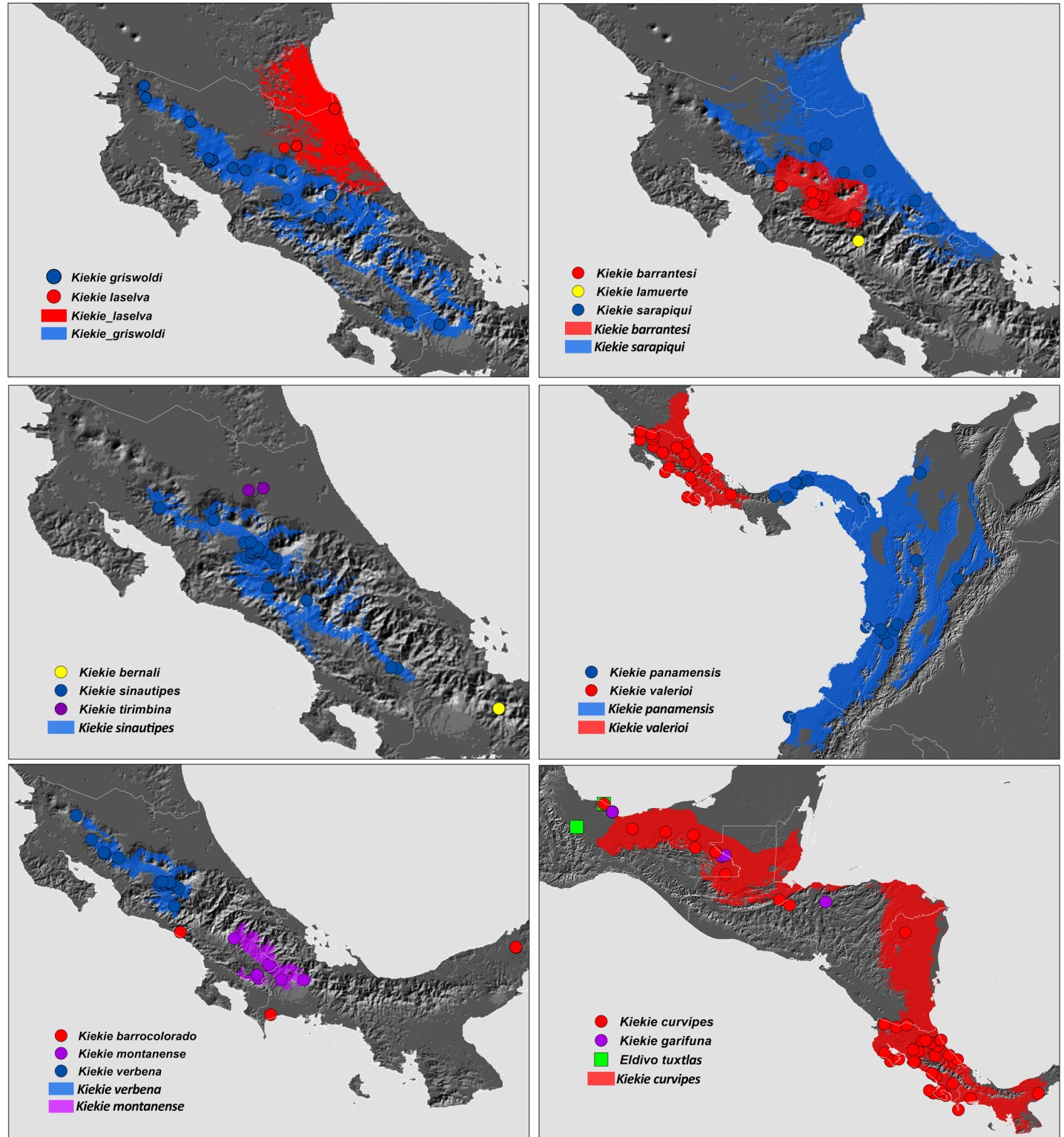

**Figure 27 Distribution map of *Kiekie* species.** Circles and colored areas represent occurrence records and distribution ranges estimated with Maxent, respectively.

16.71, metatarsus 12.83, tarsus 5.40, total 57.20; II: 15.13, 6.63, 15.39, 12.33, 5.02, total 54.50; III: 13.56, 5.50, 12.60, 12.44, 4.50, total 48.60; IV: 17.08, 6.35, 16.68, 19.14, 5.50, total 64.75. Leg spination: tibia I-II v2-2-2-2-2, metatarsus I-II v2-2-2-2-2, tibia III v-2-2-2, d1-1-1, p1-0-1, r1-0-1, metatarsus v-2-2-2, p1-1-2, r1-1-1; tibia IV v2-2-2 d1-1-1, r1-0-1, p1-0-1, metatarsus v-2-2-2, d0-1-0, p1-1-2, r1-1-2. Epigynum (Fig. 6 AD: median sector with anterior area narrow and posterior area wide, with margins sclerotized; lateral fields with a large hyaline projection, lateral projections elongated, originated medially. Vulva (Fig. 6E). Copulatory ducts elongated, curved, with a 360° turn (as in *K. sinuatipes*); spermathecae elongated with an internal projection; fertilization ducts small and located posteriorly.

**Variation.** Males (*n* = 4): Total length 27.00–34.09, carapace 14.98–19.11, femur I 18.81–20.30. Females (*n* = 6): Total length 26.89–36.50, carapace 13.76–16.22, femur I 13.90–15.16.

**Distribution.** Montane ecosystems of Costa Rica (Fig. 27).

*Kiekie barrocolorado Polotow & Brescovit, 2018*
   Figure 7.
   *Kiekie barrocolorado Polotow & Brescovit, 2018*: 367, f. 12A-B (Dm). Male holotype from Panama, Canal Zone, Barro Colorado Island (9.15, −79.85) A.M. Chickering coll., (deposited in MCZ 79093, not examined).

**Material examined.** PANAMA: Two males and one female from Chiriquí province, Puerto Amuelles, banana plantations (8.3408N, 82.8069W, 10 m), 29.VII.2019, N. Hazzi, T. Rios and J. Bernal (MCZ IZ 167571). COSTA RICA: Puntarenas province, Tres Colinas Station (9.3374N, 83.9108W), 17.IX. 2002, R. Gonzalez (MNCR).

**Diagnosis.** Males of *Kiekie barrocolorado* resemble those of *K. garifuna* by the embolus shape and conductor, but differ by the RTA position, which is distant from the apical border of the tibia (Fig. 7C) as opposed to be positioned at the apex, and by the median apophysis shape, with a narrow and longer base (Fig. 7B) instead of wide and long as most species (*Polotow & Brescovit, 2018*). Females differ from all other species in the genus by the uniquely shaped pigynal median sector (Fig. 17D), with a pointed projection in the posterior area and anterior area broad, with two lobes, and by having small and inconspicuous copulatory openings (Fig. 7E) (modified from *Polotow & Brescovit, 2018*).

**Female** (MCZ IZ 167571, Puerto Amuelles, Chiriquí province, Panama). Total length 20.30. Carapace 10.77 long and 8.72 wide. AME 0.50, ALE 0.38, PME 0.63, PLE 0.60. Sternum 4.60 long and 4.40 wide, labium 1.56 long and 1.38 wide, endites 2.89 long and 1.89 wide. Leg measurements: I: femur 8.73, 4.32, 8.06, 6.87, 2.60, total 30.58; II: 8.02, 3.72, 7.52, 6.70, 2.54, total 28.50; III: 6.57, 3.12, 6.08, 6.49, 2.30, total 24.56; IV: 8.96, 3.71, 8.72, 10.07, 3.07, total 34.53. Leg spination: tibia I-II v2-2-2-2-2, metatarsus I–II v2-2-2-2-2, tibia III v-2-2-2, d1-1-1, p1-0-1, r1-0-1, metatarsus v2-2-2, d0-1-0, p1-1-2, r1-1-2; IV-tibia v2-2-2, d1-1-1, 1-0-1, p1-0-1, metatarsus v-2-2-2, d0-1-0, p1-1-2, r1-1-2. Epigynum (Fig. 7D): median sector with a pointed or oval projection in the posterior area, anterior

area broad with two lobes, lateral fields without large hyaline projection and copulatory openings small and inconspicuous, lateral process elongated and originated medially. Vulva (Fig. 7E). Copulatory ducts curved, with a broad less sclerotized area at the beginning of the copulatory openings; spermathecae bean-shaped; fertilization ducts small and located posteriorly.

**Male.** Described by *Polotow & Brescovit (2018)*.

**Distribution.** Lowland ecosystems of Costa Rica and Panama (Fig. 27).

### *Kiekie bernali* sp. nov.
Figures 4D, 8, 23B, 25B and 27.

## Type material
*Holotype*. PANAMA: Male from Chiriquí province, Fortuna Reserve (8.711N, 82.171W, 1,800 m), 4.XII.2018, N. Hazzi (MCZ IZ 167572). *Paratype*s: Six males and five females, same data as holotype (MCZ IZ 167573).

**Diagnosis.** This is the smallest species of *Kiekie* and it differs from the other congeneric species by the following unique combination of characters: males with prominent subtegulum not covered by the base of the embolus (Figs. 8B, 23B and 25B, in the remaining species the subtegulum is visible in the exterior face) retrolateral tibial apophysis apically truncated (Fig. 8C), leaf-shaped median apophysis (Figs. 8B, 23B and 25B) and metatarsus IV unmodified. The epigynum resembles that of *K. lamuerte* sp.nov by the short copulatory ducts and small copulatory openings (Figs. 8D, 8E, 13D and 13E)), but it differs by the less sclerotized margins of the median sector and the bean-shaped spermathecae instead of kidney-shaped (Fig. 8D).

**Etymology:** This species is dedicated to the memory of the late Juan Bernal, entomologist from Universidad of Chiriquí who contributed numerous studies of the diversity and distribution of aquatic insects in Panama.

**Description.** *Male* (MCZ IZ 167572, from Fortuna Reserve, Chiriquí province, Panama). Total length 10.60. Carapace 5.90 long and 4.99 wide, eye diameters: AME 0.34, ALE 0.21, PME 0.40, PLE 0.38. Clypeal height 0.17, sternum long, wide; labium 1.65 long, 1.30 wide. Sternum 2.58 long and 2.50 wide, labium 0.54 long and 0.86 wide, endites 1.40 long and 0.88 wide. Leg measurements: I: femur 6.52, patella 1.9, tibia 7.42, metatarsus 5.49, tarsus 2.72, total 24.05; II: 6.69, 2.22, 6.49, 5.64, 2.21, total 23.25; III: 5.98, 2.00, 5.01, 5.38, 2.01, total 20.38; IV 7.58, 2.03, 7.10, 8.80, 3.18, total 28.69. Leg spination: leg I tibia v2-2-2-2-2, p0-1-0, r1-0-0, metatarsus v2-2-2, p1-1-0 r0-1-0 leg II tibia v-2-2-2-2-2, d0-0-1-0, p0-1-0, r1-1-0, metatarsus v2-2-2, p1-1-0, r1-1-0, leg III v2-1-2-2, p1-0-1-0, r1-1-1, metatarsus v2-2-2-2, d0-1-0, r1-1-1, leg IV tibia v2-2-2, d1-1-1, p-0-1-1-0, r0-1-1-0, metatarsus v2-2-2, d0-1-1, p0-1-2, r0-1-2. Palp: RTA small and truncated at the apex (Fig. 8C); embolus flagelliform and laminar with a sclerotized dark area at the base (Figs. 8B, 23B and 25B); median apophysis with a long base and horizontally oriented (Figs. 8B, 23B and 25B);

conductor with a narrow base and large apex, covering the tip of the embolus (Figs. 8B, 23B and 25B).

*Female* (MCZ IZ 167573-1, from Fortuna Reserve, Chiriquí province, Panama). Total length 12.22. Carapace 6.34 long and 4.94 wide, eye diameter: AME 0.34, ALE 0.25, PME 0.37, PLE 0.38. Clypeal height 0.15, sternum long 2.59 and 2.54 wide, endites 1.79 long and 1.04 wide, labium 0.72 long and 0.91 wide. Leg measurements: I: femur 4.87, patella 1.95, tibia 4.91, metatarsus 3.65, tarsus 1.65, total 17.03; II, 4.66, 2.14, 4.33, 3.55, 1.60, total 16.28; III 4.17, 2.03, 3.55, 3.67, 1.71, total 15.13; IV 5.52, 1.80, 4.74, 6.15, 2.46, total 20.67. Leg spination: tibia I-II v2-2-2-2-2, metatarsus I-II v2-2-2-2-2; III tibia v2-2-2, d0-1-1-0, r1-0-1, p1-0-1; metatarsus v2-2-2, p1-1-1, r1-1-1; IV tibia v2-2-2, d0-1-0-0, p0-1-1-0, r0-1-1-0; metatarsus v2-2-2-2, p1-1-1 r1-1-1. Epigynum (Fig. 8D): median sector sub quadrangular with margins sclerotized, posterior area wide and anterior area narrow, lateral process small, originated medially. Vulva (Fig. 8E). Copulatory ducts short and curved, spermathecae small and bean-shaped, fertilization ducts small and posteriorly located.

**Variation.** Males (*n* = 6): Total length 9.70–10.60, carapace 4.86–5.90, femur I 5.90–6.72. Females (*n* = 5): Total length 12.22–15.22, carapace 6.33–6.97, femur I 5.20–5.86.

**Distribution.** Montane ecosystems of Chiriquí Province, Panama (Fig. 27).

### *Kiekie curvipes* (Keyserling, 1881)

Figures 5E, 5F, 10, 11, 26E and 27.

*Microctenus curvipes* Keyserling, 1881: 579, pl. 16, fig. 24 (male holotype from Panama, deposited in C.L. Koch personal collection, not examined, probably lost).

*Ctenus curvipes* Simon, 1897:107, fig. 100; F.O. Pickard-Cambridge 1897: 86, pl. 3, figs 6a, 7e; *Polotow & Brescovit, 2014*: 355, fig. 12C.

*Ctenus incolans* F.O. Pickard-Cambridge, 1900: 111, pl. 7, figs 35–36 (female holotype from Guatemala, deposited in BMNH 1901.3.3.142–143).

*Kiekie curvipes* *Polotow & Brescovit, 2018*: 358, f. 3C, 6A-C (mf, T from *Ctenus*, S of *Ctenus incolans*).

**Material examined.** COSTA RICA. Alajuela province: one female from Upala, Finca la "Selva" (10.8977N, 85.0166W, 50 m) (UCR); one male from San Ramón, Concepción (10.1213N, 84.4402W, 1,100 m), 00.V.2015, N. Conejo (UCR); San Pedro de Poas (10.0722N, 84.2461W, 1,100 m), 15.VIII.2017, T. Roman (UCR); one female from Los Chiles, Finca de la Compañía Comercial San Luis (10.9616N. 84.6517W), 08.II. 2009 (MNCR); Guanacaste province: one female from Pitilla Biological Station (10.9926N, 85.4295W, 700 m), J. Huff (MNCR); one female from Monte Alto de Pilangosta, Hojancha Fila Maravilla (10.0085N, 85.4124W, 800 m), W. Porras (MNCR); Heredia province: one female from Sarapiquí, La Selva Biological Station (10.4306N, 84.0069W), C. Valerio (UCR); five females and five males from Sarapiquí, Tirimbina Rainforest Center (10.4150N, 84.1210W, 160 m), 15.VI.2019, N. Hazzi (MCZ IZ 167574); Limón Province: one female from Liverpool, Veragua (9.9248N, 83.1912W, 400 m), L.S egura (UCR); one

female from Santa Rosa (10.3525N, 83.8044W, 100 m), 00.04.2009, R. Madrigal (UCR); one female from Reserva Hitoy Cerere (9.6717N, 83.0277W), 28.III.2003 (MNCR); one male from Guacimo, Pocora (10.2458N, 83.5766W) 00.V.2000, C. Víquez (MNCR); four females and four males PNN Tortugueros (10.4498N, 83.4730W, 10 m), 10.VI.2019, N. Hazzi. Puntarenas province: one female from Playa Nicoya-Curu, Bosques Marianas (9.791N, 84.9281W), 3.XII.1988 (UCR); one female, San Vito, Coto Brus (8.9541N, 83.0704), C. Valerio (UCR); one female from PNN Corcovado, San Pedrillo (8.6230N, 83.7366W), 27.IX.1998, M. Lobo (MNCR); one female from Peninsula la Osa, Puerto Jiménez (8.5387N, 83.3061W), 10.I.1998, M. Lobo (MNCR); one female, Rincon Station (8.6049N, 83.5295W), 9.IX.1996; S. Avila (MNCR); Puesto en la Isla del Cano y Sendero sitio arqueológico (8.7097N, 83.8736W), 23.II.1998, A. Azofeifa; two males and two females from Cirenas (9.720N, 85.211W), 08.VI.2018, N. Hazzi (MCZ IZ 167575). San José province: one female from Puriscal, Maztatal (9.6737N, 84.3691W, 400 m), 27.III.2009, M. Ramírez (UCR); one female from Pérez de Zeledon, Escuela Naranjo (9.2035N, 83.6366W, 800 m); one female from Puriscal, PN La Cangrejera (9.7003N, 84.3978W), B. Gamboa (MNCR). PANAMA. Chiriquí province: two females and two males from Chiriqui University (8.4318N, 82.4515W, 30 m), 28.VII.2018, N. Hazzi (MCZ IZ 167576). MEXICO: Chiapas province: one female and one male from Marqués de Comillas, Playón de la Gloria (16.1513N, 90.8966W), 05.VII.2013 (CAN-AR 3462); one female and one male from Ocosingo, El Taller, Sierra de la Cojolita (16.804N, 90.901W), 09.VIII.2005, R. Paredes (CNAN-AR 6695), one female from Ocosing, Arroyo Nayte, Sierra de la Cojolita (16.792N, 91.042W), R. Paredes (CNAN-AR: 6692); Marquez de Comillas, Playon de la Gloria (16.1504N, 90.8965W), 7.VIII.2013 (CNAN-AR 4979); Veracruz Province: four females and four males, Los Tuxtlas Biological Station (18.5822N, 9.0755W), F. Alvarez Padilla (2016-2017) (CNAN-AR-Ar011669).

**Diagnosis.** Males of *Kiekie curvipes* differ from all congeneric species by the S-shaped and cylindrical RTA, medially located at the palpal tibia (Figs. 9B, 9C, 10B and 10C). Females resemble those of *K. montanense* by the absence of a hyaline area in the lateral field, close to the copulatory opening (Figs. 9D and 10D); but can be distinguished by the shorter copulatory ducts (Figs. 9E and 10E) (modified from *Polotow & Brescovit, 2018*).

**Taxonomic remarks**: *Polotow & Brescovit (2018)* mentioned the round projection at the embolus base and the bimarginated median apophysis (Fig. 6A) as diagnostic characters of *K. curvipes*. However, during examination of specimens from Mexico, we found specimens with neither the round projection at the embolus base nor the bimarginated median apophysis (Fig. 10B). In addition, we found specimens from Mexico with the round projection at the embolus base, but without the bimarginated median apophysis. Therefore, males of *Kiekie curvipes* should be diagnosed based on the S-shaped and cylindrical RTA. Females from these localities have larger spermathecae with conspicuous posterior granules (Fig. 10E) than those of the southern populations of Costa Rica and Panama. Future studies should examine in more detail whether this variation could indicate the existence of more than one species.

**Distribution.** Lowland ecosystems of Costa Rica, Panama, Mexico, Honduras and Nicaragua (Fig. 27).

### *Kiekie dietrichi* Omelko, 2023

*Kiekie dietrichi* Omelko, 2023: 280, f. 6-10, 15-18, 23-26, 31-35, 38-39. Male holotype and four female paratypes from Panama, Chiriquí Province, Totumas Mt., 8°53′3.24″N 82°40′41.05″W, deposited in ZMMU, Moscow (not examined).

**Diagnosis.** Males of *K. dietrichi* resemble those of *K. montanense* in the shape of the median apohysis (Fig. 15; Omelko, 2023: figs 15–17), but can be distinguished by the larger and curved RTA with a sharp tip (*vs.* a tiny, straight, bifurcated tip as in *K. montanense*, Fig. 15C; Omelko, 2023: fig. 17). Females of *K. dietrichi* resemble those of *K. curvipes* in the shape of the median sector of the epigynum (Figs. 9D and 10D; Omelko, 2023: figs. 31 and 32) and the orientation of the copulatory ducts (Figs. 9E and 10E; Omelko, 2023, fig. 34), but differ by the small lateral processes of the epigynum pointing towards each other and by the reniform spermathecae (Omelko, 2023: figs. 31 and 32), in contrast with the large downward-pointing lateral projections and the piriform spermathecae of *K. curvipes* (Figs. 9D and 10D) (modified from Omelko, 2023).

**Male and female.** Described by Omelko (2023).

**Distribution.** Known only from the type locality in the Chiriquí Province of Panama.

### *Kiekie garifuna* Polotow & Brescovit, 2018

Figures 11, 24C and 26E.

*Kiekie garifuna* Polotow & Brescovit, 2018: 361, f. 7A-B (Dm). Male holotype from Guatemala,deposited in BMNH 1901.3.3.142–143 (not examined).

**Material examined.** MEXICO: Veracruz province: two females and two males, Los Tuxtlas Biological Station (18.5822N, 9.0755W), F. Alvarez-Padilla (2016-2017) (CNAN-AR-Ar011670). Chiapas province: four females from Ocosingo, Arroyo Nayte, Sierra de la Cojolita (16.792N, 91.042W), A. Valdez, O. Francke, C. Santibáñez y J. Cruz (CNAN-AR:9528, 6668, 6695, 6696).

**Diagnosis.** Males of *Kiekie garifuna* can be distinguished from congeneric species by the shape of the RTA, with a wide and flat tip, and a large base (Figs. 11B and 24C). Females of *Kiekie garifuna* differ from all other the species in the genus by an ovoid projection in the anterior area of the median sector of the epigynum (Fig. 11D) and by their large and thick curved copulatory ducts (Fig. 11E).

**Female** (CNAN-AR-Ar011670, from Los Tuxtlas Biological Station, Veracruz province, Mexico). Total length 18.40. Carapace 9.00 long and 7.00 wide. AME 0.38, ALE 0.30, PME 0.45, PLE 0.48. Sternum 3.78 long and 3.38 wide, labium 1.18 long and 1.30 wide, endites 2.24 long and 1.32 wide. Leg measurements: I: femur 8.47, patella 3.66, tibia 7.85, metatarsus 6.53, tarsus 2.70, total 29.21; II: 7.81, 3.40, 7.81, 6.38, 2.45, total 27.84; III: 7.0, 2.85, 6.31, 6.15, 2.41, total 24.71; IV: 8.96, 3.71, 8.72, 10.07, 3.07, total 34.53. Leg spination:

tibia I-II v2-2-2-2-2, metatarsus I-II v2-2-2-2-2, tibia III v-2-2-2, d1-1-1, p1-0-1, r1-0-1, metatarsus v2-2-2, d0-1-0, p1-1-2, r1-1-2; IV-tibia v2-2-2, d1-1-1, 1-0-1, p1-0-1, metatarsus v-2-2-2, d0-1-0, p1-1-2, r1-1-2. Epigynum (Fig. 11D): median sector with an oval projection in the anterior area, lateral fields without large hyaline projection and copulatory openings small and inconspicuous, lateral process elongated and originated medially. Vulva (Fig. 11E). Copulatory ducts curved, with a broad less sclerotized area at the beginning of the copulatory openings; spermathecae nearly rounded; fertilization ducts small and located posteriorly.

**Male.** Described by *Polotow & Brescovit (2018)*.

**Variation.** Males (*n* = 3): Total length 14.20–15.10, carapace 8.40–10.10, femur I 9.25–10.10. Females (*n* = 3): Total length 18.22–19.00, carapace 9.10–9.55, femur I 8.34–8.90.

**Distribution.** Lowland ecosystems from Mexico to Nicaragua (Fig. 27).

*Kiekie griswoldi* Polotow & Brescovit, 2018
  Figures 5C, 5D, and 12, Fig. S2.
  *Kiekie griswoldi* Polotow & Brescovit, 2018: 365, f. 10A-D (Dmf). Male from Costa Rica, Puntarenas Province, Santa Elena, near Monteverde (deposited in MCZ 30607, not examined).
  *Kiekie sanjose* Polotow & Brescovit, 2018: 10, Fig. (Female holotype from Costa Rica, San José deposited in MCZ 79098, and female paratype deposited in AMNH, not examined).
  **New synonymy.**

**Material examined.** COSTA RICA: Alajuela Province: One female, Arenal, San Ramón Reserve (10.2276N, 84.6008W), C. Víquez (MNCR); one female, Upala, around San Ramón de dos Ríos (10.8832N, 85.4135W), 22.VI.1994 (MNCR), one male, San Ramón, Villa Blanca Cloud Forest Hotel (10.2025N, 84.4844W), 21.VII.2018, N. Hazzi and N. Conejo (MCZ IZ 167577); Cartago Province: One male, National Park Tapantí, Quebrada Segundo area (9.7625N, 83.7883W), R. Delgado (MNCR); Guanacaste province: One female, Faldas Volcán Tenorio (10.658N, 84.9961W), 08.II.1986, C. Bernal (UCR); One female, Biological Station Pitilla, 9km from Santa Cecilia (10.9926N, 85.4295W), 01. V.1989, P. Rios (MNCR); Puntarenas Province: One female, Arenal, Monteverde, La Casona Station (10.2984N, 84.7925W), 23.V.1995, K. Martínez (MNCR); One female, Arenal, Monteverde, Tropical Scientific Center (10.3014N, 84.7954W), 17.VIII.2009, D. Gutierrez (MNCR); three males and three females, Coto Brus, Las Cruces Biological Station (8.7840N, 82.9590W), 25.VI.20018, N. Hazzi and N. Conejo (MCZ IZ 167578). PANAMA: Chiriquí province: One male, Lagunas de Volcán (8.7638N, 82.6772W, 1368m), 31.VII.2018, N. Hazzi (MCZ IZ 167579).

**Diagnosis.** Males of *Kiekie griswoldi* are easily distinguished from all other species of the genus by the large and curved RTA, emerging from the retrodorsal area of the tibia to ventral direction (Figs. 12B and 12C, Figs. S2B and S2C), and by a large median apophysis

at the apex with C-shaped in ventral view (Fig. 12B, Fig. S2B). Females of *Kiekie griswoldi* differ from all species by the lateral and straight elongation of the spermathecae (fig. 12E) (modified from *Polotow & Brescovit, 2018*).

**Male.** Described by *Polotow & Brescovit (2018)*.

**Taxonomic Remarks:** *Polotow & Brescovit (2018*: fig. 10C) diagnosed females of *Kiekie griswoldi* by the short and anterior position of the lateral process of the epigynum. However, during examination of museum specimens, we discovered that females instead have lateral processes positioned medially, as usual in most species, but some specimens lack them, probably because they lost them during mating. Therefore, the lateral process that Polotow & Brescovit pointed in their fig. 10C, is not the lateral process, but is just part of the conformation of the lateral field of the epigynum. *Polotow & Brescovit (2018)* differentiate *Kiekie sanjose* from *K. griswoldi* by the longer median field of the epigynum. However, examination of specimens from areas close to the type localities and other regions in Costa Rica, indicate there is a continuum of variation between shapes in the middle field with high overlap. In addition, the males associated with females with longer median epigynal field have the diagnostic characters of *K. griswoldi*. Therefore, we synonymize *K. sanjose* with *K. griswoldi*. This species presents a large morphological variation across its geographic distribution range. Populations from the southern area of Cordillera Talamanca are considerably larger in size than the populations from the mountain ranges of Central Cordillera, Guanacaste and Tilarán. In addition, populations of the southern part of Talamanca have larger and robust median tegular projection compared to the populations of the northern area (Fig. 12 and Fig. S2). Due to the limited sampling and continuous morphological variation, we prefer to consider this variability as intraspecific. However, we acknowledge that more studies are needed to determine if *K. griswoldi* could be more than one species.

**Distribution.** Montane ecosystems of Costa Rica and Panama (Fig. 27).

*Kiekie lamuerte* **sp. nov.**
      Figures 13, 23A and 27A.

## Type material
*Holotype.* COSTA RICA: Male from San José Province, Cerro de la Muerte (9.55N, 83.72W), 21.III. 1993, G. Hormiga (MCZ IZ 167580). *Paratypes.* One male and one female, same data as holotype (MCZ IZ 167581).

**Etymology.** The species name is a toponym in apposition in reference to the type locality.

**Diagnsosis.** Males of *K. lamuerte* sp.nov resemble those of *K. bernali* sp.nov in the general palp conformation, but differ by the hook-shaped median apophysis (Figs. 13B, 23A and 25A), presence of a sclerotized process (Figs. 13B and 23A), apically bifid RTA (Fig. 13C) and the presence of modified metatarsus IV. The epigyna resemble those of *K. bernali* sp.nov by the short copulatory ducts and small copulatory openings, but differ by the
stronger sclerotized margins of the median sector (Fig. 13D) and the spermathecal shape, with a well-defined base (Fig. 13E).

**Male** (MCZ IZ 167580, from Cerro de la Muerte, San José Province, Costa Rica). Total length 18.57. Carapace 10.18 long and 8.27 wide. Eye diameters: AME 0.43, ALE 0.33, PME 0.35, PLE 0.52. Clypeal height 0.24, sternum 4.55 long and 4.31 wide, endites 2.82 long and 1.50 wide, labium 1.26 long and 1.47. Leg measurements: I: femur 10.27, patella 3.87, tibia 10.15, metatarsus 9.17, tarsus 3.78, total 37.24; II: 9.31, 3.43, 10.19, 8.24, 3.78, total 34.95; III: 8.85, 2.99, 8.37, 8.29, 3.30, total 31.80; IV: 3.22, 10.68, 11.24, 11.59, 3.67, total 40.40. Leg spination: tibia v2-2-2-2-2, p0-1-0, 0-1-0 metatarsus v2-2-2, II tibia v2-2-2-2-2, r2-2-0, p2-2-0, metatarsus v2-2-2, III v-2-2-2, d1-1-1, p 1-0-1, r1-0-1, metatarsus v2-2-2, 0-1-0, p1-1-2, r1-1-2; IV tibia v-2-2-2, d1-1-1, p1-0-1 r1-0-1, metatarsus modified. Palp: RTA small and apically bifid (Fig. 13C), embolus flagelliform and laminar, with a sclerotized process at the base of the embolus (Figs. 13B and 23A), median apophysis vertically oriented (Figs. 13B, 23A and 25A); conductor with a narrow base and large apex, covering the tip of the embolus (Figs. 13B, 23A and 25A).

**Female** (MCZ IZ 167580-1, from Cerro de la Muerte, San José Province, Costa Rica). Total length 27.76. Carapace 12.35 long and 9.35 wide. AME 0.47, ALE 0.35, PME 0.55, PLE 0.50. Clypeal height 0.26, sternum 5.14 long and 4.62 wide, endites 2.45 long and 2.36 wide, labium 1.15 long and 1.55 wide. Leg measurements: I, femur 9.31, patella 4.60, tibia 9.12, metatarsus 7.01, tarsus 2.90, total 32.93; II: 8.04, 3.47, 6.75, 6.30, 3.00, total 24.56; III: 7.51, 3.50, 6.16, 6.10, 2.30, total 25.57; IV: 9.55, 3.84, 9.10, 9.54, 3.50, total 35.53. Leg spination tibia I-II v2-2-2-2-2, metatarsus I-II v2-2-2-2-2; III tibia v2-2-2, p1-1-0, r1-1-0, metatarsus v2-2-2, d0-1-0, p1-1-1 r1-1-1; IV tibia v2-2-2, d0-1-1-0, r0-1-1-0, p0-1-1-0, metatarsus v2-2-2, p1-1-1, r1-1-1, r1-1-1-2. Epigynum (Fig. 13D): median sector sub quadrangular with margins sclerotized, posterior area wide and anterior area narrow, lateral process small, originated medially. Vulva (Fig. 13E). Copulatory ducts short and curves, spermathecae bean-shaped, with a well-defined base; fertilization ducts small, posteriorly located.

**Variation.** Males (*n* = 2): Total length 18.57–19.32, carapace 10.75–10.90, femur I 10.27–10.75.

**Distribution.** Montane ecosystems of San José Province, Costa Rica (Fig. 27).

*Kiekie laselva* **sp. nov.**
    Figures 4E, 4F, 24A, 26A and 27.

## Type material

*Holotype*. COSTA RICA: Female from Heredia Province, Sarapiquí, La Selva Biological Station (10.4305N, 84.0073W), 18.VI.2018, N. Hazzi (MCZ IZ 167582). *Paratypes*: Three males and five females, Limón Province, PNN Tortugueros (10.4498N, 83.4730W, 10 m), 10,VI.2019, N. Hazzi (MCZ IZ 167583); One female, Heredia Province, Sarapiquí, Tirimbina Rainforest Center (10.4150N, 84.1210W, 160 m), 25.VI.2018, N. Hazzi (MCZ IZ 167584).

*Other material examined.* **Costa Rica:** One female, Limón Province, River Fri (10.4014N, −83.6000W), 04.X.1997, C. Víquez (MNCR).

**Etymology.** The species name is a noun in apposition taken from La Selva Biological station, which was established as a research station in 1968 by the Organization of Tropical Studies (OTS).

**Diagnosis.** Males of *K. laselva* sp.nov resemble those of *K. valerioi* sp.nov and *K. panamensis* by the general conformation of the palp, but can be distinguished from them by a white conspicuous laminar process of the embolus (Figs. 14A and 24A) and by a longer whip-shaped embolus (Figs. 14A, 24A and 26A). Females of *K. laselva* sp.nov also resemble those of *K. panamensis* and *K. valerioi* sp.nov in the general epigynal configuration but differ by the more complex coiled conformation of the copulatory ducts (Fig. 14E).

*Female* (MCZ IZ 167582, from La Selva Biological Station, Sarapiquí, Heredia Province, Costa Rica). Total length 37.12. Carapace 16.57 long and 13.22 wide. AME 0.62, ALE 0.56, PME 0.69, PLE 0.82. Clypeal height 0.26, sternum 6.70 long and 6.03 wide, endites 4.00 long and 2.66 wide. Leg measurements: I: femur 14.19, patella 6.00, tibia 13.88, metatarsus 9.80, tarsus 4.06, total 48.53; II: 13.10, 5.55, 11.82, 9.86, 3.60, total 43.93; III: 11.49, 4.71, 9.98, 9.05, 2.90, total 38.13; IV: 14.05, 5.83, 12.78, 14.82, 4.00, total 51.48. Leg spination: tibia I-II v2-2-2-2-2, metatarsus I-II v2-2-2-2-2, tibia III v-2-2-2, d1-1-1, r1-0-1, p1-0-1, metatarsus v-2-2-2, d0-1-0, p1-1-2, r1-1-2; IV tibia v2-2-2 d1-1-1, r1-0-1, p1-0-1, metatarsus v-2-2-2, d0-1-0, p1-1-2, r1-1-2. Epigynum (Fig. 14D): median sector sub pentagonal and vertically elongated, lateral fields with a large hyaline projection, posterior area wide and anterior area narrow, lateral process sharp, originated medially. Vulva (Fig. 14E). Copulatory ducts elongated forming complex loops, spermathecae small and bean-shaped; fertilization ducts small and located posteriorly.

**Male** (MCZ IZ 167583-1, La Selva Biological Station, Sarapiquí, Heredia Province, Costa Rica). Total length 36.84. Carapace 20.02 long and 15.80 wide. AME 0.65, ALE 0.50, PME 0.74, PLE 0.86. Clypeal height 0.20, sternum 8.81 long and 7.36 wide, labium 2.44 long and 2.24, endites 4.48 long and 2.60 wide. Leg measurements: I: femur 19.84, patella 7.18, tibia 19.55, metatarsus 17.74, tarsus 6.69, total 71.00; II: 18.40, 7.38, 17.52, 15.29, 5.30, total 63.89; III: 16.64, 6.3, 14.64, 13.45, 4.60. total 55.61; IV: 20.80, 8.05, 20.10, 20.33, 6.38, total 75.60. Leg spination: leg I tibia v2-2-2-2-2, p1-1-1-0, r1-0-0, metatarsus v-2-2-2, Leg II, tibia v2-2-2-2, p-1-1, r1-1, metatarsus v-2-2-2; III: v2-2-2, d1-1-1, p 1-0-1, r1-0-1, metatarsus v2-2-2, d0-1-0, p1-1-2, r1-1-2, IV tibia v2-2-2, d1-1-1, p1-01, r1-0-1, metatarsus modified. Palp: RTA spiniform in ventral view, with wide apex (Figs. 14B and 14C), embolus elongated, flagelliform with a large white laminar process, presence of sclerotized process at the base of the embolus (Figs. 14B and 24A), median apophysis with cup-shaped aperture visible ventrally (Figs. 24A and 26A); conductor with a narrow base and large apex, covering the tip of the embolus (Figs. 24A and 26A).

**Variation.** Males (*n* = 3): Total length 36.50–37.68, carapace 19.00–20.68, femur I 19.00–20.51. Females (*n* = 7): Total length 34.53–40.32, carapace 17.63–20.01, femur I 14.19–17.6.

**Distribution.** Lowland ecosystems of Costa Rica (Fig. 27).

***Kiekie montanense* Polotow & Brescovit, 2018.**
   Figures 5B and 15.
   *Kiekie montanense* 364, f. 9A-D (Dmf). Male holotype from Costa Rica, Puntarenas Province, San Vito (8.83N, 82.92W) (deposited in CAS, not examined).

**Material examined.** PANAMA: 10 females and 10 males from Panama, Chiriquí province, Lagunas de Volcán (8.7638N, 82.6772W, 1,368 m), 31.VII.2018, N. Hazzi (MCZ IZ 167585); COSTA RICA: 3 males and 3 females from Puntarenas province, Las Cruces Biological Station (8.7840N, 82.9590W, 1,200 m), 25.VI.2018, N. Hazzi (MCZ IZ 167586); 3 males and 1 female, Puntarenas Province, Las Alturas Biological Station (8.9450, 82.8330W, 1,300 m), 29.VI.2018, N. Hazzi (MCZ IZ 167587). COSTA RICA: Puntarenas Province: One female, Buenos Aires. Durika Reserve (9.2617N, 83.2469W), 00.II.1990, one male, Coto Brus, Sabalito, Finca Marco Morales (8.9064N, 82.8047W), 08.V.1995 (MNCR); one male, Coto Brus, 500m from Cerro Pelon (8.9136N, 82.7965W) (MNCR).

**Diagnosis.** Males of *K. montanense* resemble those *of K. curvipes* and *K. verbena* by having a prominent tegular process (Figs. 9B, 15B and 21B), but can be distinguished from them by a median apophysis with a long and straight stem and a small RTA (Figs. 15B and 15C), not visible in ventral view. Females differ from all other congeneric species by the unique configuration of the copulatory ducts (Fig. 15E), which are coiled and connecting posteriorly (below) with the spermathecae, and by the presence of numerous granules under the copulatory ducts (modified from *Polotow & Brescovit, 2018*).

**Female** (MCZ IZ 167585-1, Lagunas de Volcán, Chiriquí province, Panama). Total length 24.00. Carapace 11.19 long and 9.18 wide. AME 0.52, ALE 0.38, PME 0.59, PLE 0.60. Sternum 4.73 long and 4.31 wide, labium 1.51 long and 1.48 wide, endites 3.07 long and 1.73 wide. Leg measurements: I: femur 8.96, patella 4.20, tibia 8.74, metatarsus 6.52, tarsus 2.80, total 31.22; II: 8.82, 3.99, 7.70, 6.52, 2.70, total 29.73; III: 7.80, 3.23, 6.54, 6.26, 2.87, total 26.70; IV: 9.58, 4.02, 9.28, 9.63, 3.00, total 35.51. Leg spination: tibia I-II v2-2-2-2-2, metatarsus I-II v2-2-2-2-2, tibia III v-2-2-2, d1-1-1, p1-0-1, r1-0-1, metatarsus v2-2-2, d0-1-0, p1-1-2, r1-1-2; IV-tibia v2-2-2, d1-1-1, 1-0-1, p1-0-1, metatarsus v-2-2-2, d0-1-0, p1-1-2, r1-1-2. Epigynum (Fig. 15D): median sector sub pentagonal, anterior area narrow, lateral fields without large hyaline projection, lateral process robust and originated medially. Vulva (Fig. 15E). Copulatory ducts elongated, coiled and connecting posteriorly (below) of the spermathecae; rounded spermathecae; fertilization ducts small, posteriorly located, with multiple granules between the copulatory and fertilization ducts.

**Male.** Described by *Polotow & Brescovit (2018)*.

**Variation.** Males (*n* = 10): Total length 20.08–21.53, carapace 10.87–11.85, femur I 10.67–11.03. Females (*n* = 10): Total length 19.39–25.57, carapace 10.42–11.88, femur I 8.04–9.54.

**Distribution.** Montane ecosystems of Costa Rica and Panama (Fig. 27).

### *Kiekie panamensis* Polotow & Brescovit, 2018
Figures 4A–4C, 16, 23E, 25E and 27.

*Kiekie panamensis* Polotow & Brescovit, 2018: 368, f. 13A-D (Dmf). Male holotype from Panama, Canal Zone, Barro Colorado Island (9.15, −79.85), deposited in MCZ 79100, (not examined).

**Material examined.** PANAMA: Coclé Province: One female, 50 km west of Penonome *Via* el Copé (8.6680N, 80.5925W), 08.VI.2008, G. Hormiga (MCZ IZ 167588); three females and three males, Colón Province, Gamboa (9.0980N, 79.0738W), 05.VIII.2018, N. Hazzi and S. Meneses (MCZ IZ 167589). COLOMBIA: Antioquia Department: one female and one male, Valle del Aburra (6.2375N, 75. 5804W), N. Hazzi and A. Arroyave (MUSENUV); Valle del Cauca Department: One female, Tulua, Botanical Garden INCIVA (4.0396N, 76.1683W), 00.X.2013, N. Hazzi (MUSENUV); one female and one male; Buga, Parque Natural Regional "El Vínculo" (3.8373N, 76.3002W), 00.X.2013, N. Hazzi (MUSENUV); one male, Cali, Universidad del Valle Melendez, Biological Station (3.3712N, 76.5331W), 00.X.2010, N. Hazzi (MUSENUV); one female and one male, Buenaventura, Pericos River Forest Reserve (3.8443N, 76.7901W), 04.I.2014, N. Hazzi (MUSENUV); one female, Buenaventura, Bahía Chucheros (3.9369N, 77.2967W), 00. XI.2011, N. Hazzi (MUSENUV). ECUADOR: Esmeraldas, Caimito (3.9369N, 77.2967W), 05.X.2019, N. Hazzi (MCZ IZ 167589).

**Diagnosis.** Males of *K. panamensis* resemble those of *K. valerioi* sp.nov but can be distinguished from them by a shorter, sclerotized tegular projection at the base of the median apophysis (Figs. 16B, 23E and 25E). Females of *K. panamensis* can be distinguished from *K. valerioi* sp.nov by sharper lateral process of the epigynum (Fig. 16D), anterior area of the median sector of the epigynum wide (Fig. 16D); and small spermathecae (16E) (modified from *Polotow & Brescovit, 2018*).

**Male and female**. Described by *Polotow & Brescovit (2018)*.

**Distribution.** Lowland and premontane ecosystems of Panama, Colombia and Ecuador (Fig. 27).

### *Kiekie sarapiqui* Polotow & Brescovit, 2018
Figures 17, 24D, 26D and 27.

*Kiekie sarapiqui* Polotow & Brescovit, 2018: 362, f. 3A, 8A-D (Dmf). Male from Sarapiquí Heredia province, Costa Rica (10.45, 84.01W), deposited in MCZ 79099 (not examined).

**Material examined.** COSTA RICA. Heredia province: two males and two females from Sarapiquí, Tirimbina Rainforest Center (10.4150N, 84.1210W, 160 m), 15.VI.2019, N. Hazzi (UCR). Alajuela province: one male and one female from Reserva Alberto Manuel Brenes (10.2301N, 84.6271W, 700 m), 00.IV.2009, A. Rojas (UCR); Limon province: one female from Tortugueros, Campus Escuela Agricola Regional del Trópico Húmedo (10.20N, 83.6167W, 50 m), 18.IX.2000, C. Víquez (MNCR); one female from Pococi, Finca del INBIO (10.1867N, 83.8569W, 300 m) (MNRC); Puntarenas province: one female from R.B. Hitoy Cerere (9.6650N, 83.0271W, 250 m) (MNCR).

**Diagnosis.** Males of *Kiekie sarapiqui* resemble those of *K. sinuatipes* by the conformation of the RTA and embolus but differ by the wider and shorter median apophysis with the aperture not visible in ventral view, and a narrower conductor (Figs. 17B, 24D, 26D). Females of *Kiekie sarapiqui* resemble those of *K. sinuatipes* but differ by a longer median epigynal field and shorter lateral process (Fig. 17D); and by copulatory ducts folded over themselves before connecting to the spermathecae (Fig. 17E).

**Male and female**. Described by *Polotow & Brescovit (2018)*.

**Distribution.** Lowland ecosystems of Costa Rica (Fig. 27).

### *Kiekie sinuatipes* (F.O. Pickard-Cambridge, 1897)
Figures 18, 24F and 26F.

*Ctenus sinautipes* F.O. Pickard-Cambridge, 1897. 84, pl. 3, f. 1b, 4d, 6e, 7f (Dmf). Male from San José Province, Costa Rica (deposited in BMNH 1896.3.20.26–29, not examined).

*Kiekie sinuatipes* *Polotow & Brescovit, 2018*: 355, f. 1A-B, 2A-F, 3B, 4A-F, 5A-C (mf, T from *Ctenus*).

**Material examined.** COSTA RICA: Alajuela province: one male from Zarcero (10.1834N, 84.3927W, 1,700 m), 18.IV.2013, A. del Valle (UCR), one male from the same locality, 27.10.1997 (MNCR). Cartago Province: One male from Dulce Nombre (9.8456N, 83.9095W), 02.IV.1978 (UCR); one female from San Rafael, Sierras de la Unión (9.9084N, 83.9719W, 1,400 m), 06.09.2017, M. Springer (UCR); one female from Alto de Ochomogo (9.8878N, 83.9413W, 1,500 m) (UCR). Heredia Province: one male from San Rafael (10.0116N, 84.1035W), 6.19.2019 (MNCR); one female from San Joaquín de Florez (10.0062N, 84.1537W), 01.05.1997, C. Víquez (1997). Puntareas province: one female fron Monteverde (10.2749N, 84.8255W), C. Valerio (UCR); one female from Coto Brus, Estación Pitter, ''Cerro Pitter'' trail (9.0256N, 82.9627W, 1,670 m), E. Núñez (MNCR), two females from the same locality, 16.01.2000 (MNCR); one female from Buenos Aires, Pila Sector Altamira (9.0329N, 83.0109W, 1,400 m), 25.VIII.2005 (MNCR). San José province: one male from Curridabat, Granadilla (9.9221, 84.0177W), C. Valerio (UCR); one male from Alajuelita, San Juan de Dios de Desamparados (9.9013N, 84.0995W), 08.IV.1981 (UCR); one male and one female from Montes de Oca (9.9407N, 84.0250W), 08.V.1981, A. del Valle (UCR), one female from San Antonio de Alajuelita (9.8858N, 84.1144W), 10. VII.1986, M. Garcia (UCR); one female from San Rafael de Moravia (9.9688N, 84.0489W), 00.V.1963, C. Valerio (UCR); one male from Guadalupe (9.9471N, 84.0535W), 06.II.1972

(S. Salas); one male from San José, Barrio Cuba (9.9245N, 84.0901W), (UCR); one female from Tibas (9.9576N, 84.0816W, 1,200 m), 00.III.1998, L. Bonatti (UCR); Estación Cuericí, 4.6 km E de Villa Mills (9.5552N, 83.6703W), A. Mora (MNCR); two males from Curridabat, Los Prados (9.9339N, 84.0284W), 10.II.1995 (MNCR); San Antonio de Escazú, William Eberhard and Mary Jane Eberhard house (9.8975N, 84.1377W), 20.VII.2018, N. Hazzi (MCZ IZ 167590).

**Diagnosis.** Males of *Kiekie sinuatipes* resemble those of *K. sarapiqui* (Figs. 18A and 18B) by the shape of the embolus and RTA but can be distinguished by the elongated hook-shaped median apophysis (Figs. 18B, 24F, and 26F). Females resemble those of *K. barrantesi* sp.nov by the conformation of the copulatory ducts (Fig. 18E) but can be distinguished from them by the narrower anterior area of the median sector narrow (Fig. 18D) and rounded spermathecae without an internal projection (Fig. 18E) (modified from *Polotow & Brescovit, 2018*).

**Male and female.** Described by *Polotow & Brescovit (2018)*.

**Distribution.** Montane ecosystems of Costa Rica (Fig. 27).

*Kiekie tirimbina* **sp. nov.**
  Figures 19, 23C, 25C and 27.

## Type material

*Holotype.* COSTA RICA: Male from Heredia Province, Sarapiquí, Tirimbina Rainforest Center (10.4150N, 84.1210W, 160 m), 15.VI.2019, N. Hazzi (MCZ IZ 167591).

**Diagnosis.** Males of *K. tirimbina* sp.nov differ from the remaining species of the genus by the combination of the following unique characters: thin and straight median apophysis, rounded at the apex (Figs. 19B, 23C and 25C); conductor large and deeply folded (Figs. 19B, 23C and 25C), tegulum with numerous and small grooves (Fig. 25C).

**Etymology.** The species epithet is a noun in apposition taken from the Tirimbina Biological Reserve, a protected tropical rainforest in Sarapiquí.

**Male** (MCZ IZ 167591, from Tirimbina Rainforest Center, Sarapiquí, Heredia province). Total length 19.12. Carapace 9.88 long and 7.91. AME 0.57, ALE 0.31, PME 0.59, PLE 0.62, sternum 4.12 long and 3.75, labium 1.40 long and 1.25 wide, endites 2.82 long and 1.41. Leg measurements: I: femur 9.91, patella 4.45, tibia 12.05, metatarsus 9.72, tarsus 3.80, total 37.93; II: 9.84, 4.24, 10.81, 9.77, 3.30, total 37.96; III: 9.54, 3.48, 8.39, 8.74, 2.81, total 32.96; IV: 11.51, 3.91, 11.13, 13.99, 4.18, total 44.72. leg I tibia v2-2-2-2-2, p1-1-0, r1-1-0, metatarsus v2-2-2; p1-1-0, r1-1-0; II-tibia v2-2-2-2-2, d1-1-1, p1-1-0, r1-1-0, metatarsus v2-2-2, p1-1-0, r1-1-0; III-tibia v2-2-2, d1-1-1, p1-0-1, r1-0-1, metatarsus v2-2-2, d0-1-0, p1-1-1, r1-1-1; IV-tibia v2-2-2, d1-1-1, p1-01, r1-0-1, metatarsus modified. Palp: RTA short and robust (Fig. 19C), embolus elongated and flagelliform, with laminar process (Figs. 23C and 25C); presence of sclerotized process at the base of the embolus (Figs. 23C and 25C); median apophysis thin and straight, apically rounded, cup-shaped with aperture

not visible ventrally (Figs. 19B, 23C and 25C); conductor with a narrow base and large apex, covering the tip of the embolus (Figs. 23C and 25C).

**Female.** Unknown.

**Distribution.** Lowland ecosystems of Costa Rica (Fig. 27).

*Kiekie valerioi* **sp. nov.**
Figures 20, 23F, 25F and 27.

## Type material

*Holotype.* COSTA RICA: Male from Heredia Province, Sarapiquí, Tirimbina Rainforest Center (10.4150N, 84.1210W, 160 m), N. Hazzi (MCZ IZ 167592). *Paratypes.* Five males and five females same data as holotype (MCZ IZ 167593).

*Additional material examined.* Costa Rica: San José Province: One male, Perez Zeledón, San Isidro del General (9.3547N, −83.6348), 3.VI.1972, J. Bolenige (UCR); One female, Puriscal, PN la Cangreja (9.7003N, 84.3978W), B. Gamboa (MNCR); Two females, Alajueja Province, San Ramón, Reserva Alberto Manuel Brenes (10.2301, 84.6271W), 15. V.2013, M. Salazar; Upala. Finca "La Selva" (10.8977N, −85.0166) (UCR); One female, Upala, El Pilón (10.8166N, −84.9500W), 16.X.2003 (MNCR), One female (10.7046N, 84.9923W), 23.VIII.2009, A. Azofeida (MNCR); Cartago Province, Turrialba (9.9067N, 83.6801W), 03.VII.1986, C. Valerio (UCR); Jimènez, Pejibaye, Copal Biological Station (9.1963N, 83.5975W), R. González (MNCR); Guanacaste Province: One male Arenal, Tilaran (10.4563N, 84.9713W), C. Valerio; Liberia, Colorado river (10.6842N, 85.4430W) (UCR); one female and one male, La Cruz, Pitilla Biological Estation (10.9926N, 85.4295W), 31.VII. 1991, C. Moraga (MNCR); Heredia Province, Sarapiquí, Tirimbina Biological Station (10.4190N, 84.1210W), (MNCR); 2 females, Limón Province, Hitoy Cerere Biological Reserve (9.6717N, 83.0277W), 14.V.1998 (MNCR); Cocori (10.5942N, 83.7165W), 00.XI.1997, E. Rojas (MNCR); Limón, Espavel trail (9.6680N, 83.0236W), 25. IX.2003 (MNCR); Pococi, Finca INBIO (10.1867N, 83.8569W) (MNCR); Puntarenas Province: San Pedrillo Station, Osa Peninsula (8.6230N, 83.7366W) (MNCR); Parrita (9.5230N, 84.5387W) (MNCR).

**Diagnosis.** Males of *K. valerioi* sp.nov resemble those of *K. panamensis* in the general palp conformation, but can be distinguished from them by a wider tegular process at the base of the median apophysis (Fig. 20B), which is not sclerotized (Fig. 23F); and a wide tegular projection at the base of the median apophysis (Fig. 25F). Females of *K. valerioi* sp.nov can be distinguished from *K. panamensis* by less sharp lateral process of the epigynum (Fig. 20D), more narrow anterior area of the median sector of the epigynum (Fig. 20D); and larger spermathecae projecting internally and contacting with the copulatory ducts (Fig. 20E), while in *K. panamensis* the spermathecae do not contact the copulatory ducts.

**Etymology.** This species is dedicated to Carlos E. Valerio, who made important contributions to the knowledge of spider diversity of Costa Rica.

**Male** (MCZ IZ 167592, from Tirimbina Rainforest Center, Sarapiquí, Heredia province). Total length 31.55. Carapace 17.50 long and 14.95 wide. AME 0.77, ALE 0.51, PME 0.81, PLE 0.87. Sternum 7.87 long and 7.01 wide, labium 2.40 long and 2.23 wide, endites 4.99 long and 2.41 wide. Leg measurements: I: femur 19.10, patella 6.86, tibia 17.99, metatarsus 15.03, tarsus 5.92, total 64.90; II: 17.37, 6.43, 18.10, 15.16, 5.06, total 62.12; III: 16.32, 6.82, 14.63, 14.44, 4.62, total 56.83; IV: 19.86, 7.63, 19.33, 20.61, 6.62, total 74.05. Leg spination: leg I tibia v2-2-2-2-2, d1-0-1, p1-1-0, r1-1-0, metatarsus v2-2-2; p1-1-0, r1-1-0; II-tibia v2-2-2-2-2, d1-1-1, p1-1-0, r1-1-0, metatarsus v2-2-2, p1-1-0, r1-1-0; III-tibia v2-2-2, d1-1-1, p1-0-1, r1-0-1, metatarsus v2-2-2, d0-1-0, p1-1-1, r1-1-1; IV-tibia v2-2-2, d1-1-1, p1-01, r1-0-1, metatarsus modified. Palp: RTA spiniform (ventral view) and with wide apex (Fig. 20C), embolus elongated and flagelliform with laminar process (Figs. 23F, 25F), with a sclerotized process at the base of the embolus (Figs. 23F, 25F); median apophysis with cup-shaped aperture visible ventrally (Figs. 20B, 23F, 25F); conductor with a narrow base and large apex, covering the tip of the embolus (Figs. 20B, 23F, 25F).

**Female** (MCZ IZ 167593-1, from Tirimbina Rainforest Center, Sarapiquí, Heredia province). Total length 34.76, Carapace 17.30 long and 15.23 wide. AME 0.74, ALE 0.56, PME 0.87, PLE 0.90. Sternum 7.50 long and 6.76 wide; labium 2.36 long and 2.26 wide, endites 4.30 long and 2.91 wide. Leg measurements: I: femur 16.45, patella 7.63, tibia 16.90, metatarsus 13.58, tarsus 4.57, total 59.13; II: 15.94, 7.60, 15.35, 12.36, 4.57, total 55.82; III: 12.79, 6.35, 11.29, 11.37, 4.27, total 46.07; IV: 17.29, 6.77, 15.85, 18.73, 5.38, total 64.02. Leg spination: tibia I-II v2-2-2-2-2, metatarsus I-II v2-2-2-2-2, tibia III v-2-2-2, d1-1-1, p1-0-1, r1-0-1, metatarsus v2-2-2, d0-1-0, p1-1-2, r1-1-2; IV-tibia v2-2-2, d1-1-1, 1-0-1, p1-0-1, metatarsus v-2-2-2, d0-1-0, p1-1-2, r1-1-2. Epigynum (Fig. 20D): median sector sub pentagonal, vertically elongated and with anterior area narrowed, lateral fields with a large hyaline projection, lateral process originated medially. Copulatory ducts elongated, with one loop and folded; large spermathecae projecting internally and contacting copulatory ducts; fertilization ducts small and posteriorly located (Fig. 20E).

**Variation.** Males (*n* = 6): Total length 25.00–36.74, carapace 14.50–21.00, femur I 18.81–20.30. Females (*n* = 6): Total length 26.44–37.03, carapace 12.56–18.14, femur I 10.78–16.45.

**Distribution.** Lowland ecosystems of Costa Rica (Fig. 27).

### *Kiekie verbena* Polotow & Brescovit, 2018
Figures 21, 23D, 25D and 27.

*Kiekie verbena* Polotow & Brescovit, 2018: 367, f. 11C-D (Df). Female from Costa Rica, San José Province, San José, La Verbena (9.65N, 83.96W), deposited in MCZ 79102 (not examined).

*Material examined.* COSTA RICA: Two females and five males from San José province, San Pedro, campus of Universidad of Costa Rica (9.9376N, 84.0507W, 1,200 m), 10.VI-2018, N. Hazzi (MCZ IZ 167594); Three males and three females, San Antonio de Escazu,

San José province (9.8975N, 84.1377W, 1,330 m), N. Hazzi, (MCZ IZ 167595); three females from Punta Arenas, Monteverde, UGA, (10.2829, 84.799W, 1,100 m), 12.06.2018, N. Hazzi (MCZ IZ 167596). Alajuela Province, San Ramón, Alberto Manuel Brenes Reserve (10.2283N, 84.6396W) (UCR); Cartago Province: Tres Rios, Concepcion (9.8638N, 83.9161W) 1981, C. Valerio (UCR); one female, La Union, San Rafael, Sierras la Unión (9.9083N, 83.9764W), 16.VIII.2015, M. Springer (UCR); Guanacaste Province: Tilaran (10.4563N, 84.9713W), 15.VII. 1967, C. Valerio (UCR); Volcán Miravalles (10.7471N, 85.1512W), 19.III.1967 (UCR); Puntarenas Province: Monteverde, San Luis (10.2852N, 84.8199W) 00.XI.1993 (MNCR). San José Province: One female, San Pedro, Universidad of Costa Rica (9.9369N, 84.0510W), 20.V.2009, R. Quesada (UCR); one female, Escazú, Agres River (9.9369N, 84.0510W), 14.III.2009, R. Arias (UCR); one female, Curridabat, Granadilla (9.9221N, 84.0177W), C. Valerio (UCR); one female, Montes de Oca (9.9407N, 84.0250W), 21.IV.1967, C. Valerio (UCR); two males, Curridabat, Los Prados (9.9339N, 84.0284W), 10.II.1995 (MNCR).

**Diagnosis.** Males of *K. verbena* resemble those of *K. curvipes* and *K. montanense* in having a prominent tegular process (Fig. 21B), but can be distinguished from them by a not strongly sclerotized tegular process, whitish in coloration. In addition, males have a bifid RTA at the base, forming two divergent apophyses (Figs. 21B and 21C), a short embolus and a small conductor (Figs. 21B, 23D and 25D) in contrast with all other congeneric species. Females are distinguished from all other species by the configuration of the epigynum with a short and wide median sector, long lateral projection of the lateral field and short copulatory openings (Fig. 21D) and short copulatory ducts (Fig. 21E).

**Male** (MCZ IZ 167594-1, from Campus of Universidad of Costa Rica, San Pedro, San José province, Costa Rica). Total length 15.09. Carapace 8.48 long and 7.00 wide. AME 0.40, ALE 0.25, PME 0.40, PLE 0.41. Sternum 3.70 long and 3.38 wide, labium 0.95 long and 1.08, endites 1.92 long and 1.03 wide. Leg measurements: I: femur 8.05, patella 2.77, tibia 7.43, metatarsus 6.66, tarsus 2.88, total 27.78; II: 7.62, 2.90, 7.10, 6.26, 2.43, total 26.31; III: 6.93, 2.92, 5.65, 6.15, 2.59, total 24.24; IV: 8.36, 2.91, 8.28, 8.75, 3.28, total 31.58.
Leg spination: leg I tibia v2-2-2-2-2, p1-1-0, r1-1-0, metatarsus v2-2-2; d1-0-1, p1-1-0, r1-1-0; II-tibia v2-2-2-2-2, d0-1-0, p1-1-0, r1-1-0, metatarsus v2-2-2, p1-1-0, r1-1-0; III-tibia v2-2-2, d1-1-1, p1-0-1, r1-0-1, metatarsus v2-2-2, d0-1-0, p1-1-1, r1-1-1; IV-tibia v2-2-2, d1-1-1, p1-01, r1-0-1, metatarsus modified. Palp: RTA at the base forming two divergent apophyses (Fig. 21B), embolus short without laminar process (Fig. 23D); embolus base without a sclerotized process (Fig. 23D); median apophysis large with cup-shaped aperture visible ventrally (Figs. 23D, 25D); conductor reduced, covering the tip of the embolus (Figs. 23D, 25D).

**Female.** Described by *Polotow & Brescovit (2018)*.

**Variation.** Males (*n* = 9): Total length 11.83–16.03, carapace 6.47–9.78, femur I 6.65–9.22. Females (*n* = 6): Total length 15.10–21.08, carapace 7.44–10.26, femur I 5.51–8.05.

**Distribution.** Montane ecosystems of Costa Rica (Fig. 27).

**Female.** Described by *Polotow & Brescovit (2018)*.

**_Eldivo_ gen. nov.**
Figures 22, 26G–26I and 27.

**Type species:** *Eldivo tuxtlas* sp.nov.

## Type material

Holotype. MEXICO: Male from Veracruz Province, Los Tuxtlas Biological Station (18.5822N, 9.0755W), F. Álvarez Padilla (2016-2017) (CNAN-AR-T01865). *Paratypes.* Three females, same data as holotype (CNAN-AR-T01866).

*Other Material Examined.* Mexico: Oaxaca province: One female, Huautla de Jiménez, San Agustín, Zaragosa (18.11N, 96.80W), 13.X.2014, O. Francke and J. Cruz (CNAN-AR). One male, San José, Nuevo Rio Manzo, cerro Chango (17.706N, 96.899W), 05.IX.2018 (CNAN-AR).

**Etymology.** This species is dedicated to the memory of Alberto Aguilera Valadez, known professionally as Juan Gabriel and colloquially as "El Divo". Juan Gabriel was a Mexican singer and songwriter that was known for his histrionic style, which overcame barriers within the Latin music.

**Diagnosis.** Males of *Eldivo* resemble those of *Kiekie* by the presence of a modified metatarsus IV, but can be distinguished from them by the shorter embolus with a laminar fold in the internal side (Figs. 22B, 26H), locking lobes located at posterior prolateral side (Fig. 26G); in contrast with the elongated embolus and locking lobes in the posterior retrolateral side in *Kiekie*. Females cannot be accurately distinguished morphologically from *Kiekie* or other Mesoamerican ctenids.

**Description.** Large-sized ecribellate spiders, total length 24.00–27.00. Carapace piriform, dark brown, black pigment around eyes; thoracic groove longitudinal, in the posterior third. Ctenid eye pattern 2–4-2, with anterior and posterior rows recurved in dorsal view. Eyes round, except oval anterior lateral eyes, with white setae around PLE and PME and lateral of AME. Chilum divided. Clypeus with long erect black bristles. Chelicerae brown, retromargin with four similar-sized teeth and two small proximal teeth; with intermarginal denticles between the promargin and the retromargin; prominent basal condyle. Endites brown with anterolateral serrula and anteromedian scopulae. Labium dark brown, slightly longer than wide. Sternum truncated anteriorly and posteriorly pointing, not extending between coxae IV. Ventral faces of coxae light brown with dark brown spots. Male legs longer and slender than female legs. Trochanter notched. Tarsus with claw tufts composed of tenent setae. Abdomen oval, brown, dorsum with a black anterior border, lighter centrally and with a folium-like pale brown longitudinal band, venter brown with four divergent series of white dots. Spinnerets: anterior laterals (ALS) long and wide in the apex; posterior medians (PMS) roughly conical, short and narrow in the apex; posterior laterals

(PLS) long and conical. Palp: RTA short and truncated in the apex (Fig. 22C), embolus short without laminar process (Figs. 22B and 26I); lacking a sclerotized process at the base (Figs. 22B and 26I); median apophysis large with cup-shaped aperture ventrally visible (Figs. 22B, 26G–26I); conductor reduced, covering the embolus tip (Figs. 22B, 26H). Epigynum (Fig. 22D): median sector subquadrangular, anterior area broad and elevated, lateral fields without large hyaline projection and copulatory openings small and inconspicuous, lateral process elongated and originated medially. Copulatory ducts curved, with a broad, less sclerotized area at the beginning of the copulatory openings; spermathecae reniform-shaped; fertilization ducts small and posteriorly located (Fig. 22E).

Composition: Only the type species, *Eldivo tuxtlas* sp. nov.

### *Eldivo tuxtlas* sp.nov.

**Etymology.** The species epithet is a noun in apposition after the type locality, Reserva de la Biosfera Los Tuxtlas, a protected tropical rainforest in Veracruz.

**Diagnosis.** As in genus description for males. Females can be recognized from other Central and North American ctenids by having a median sector subquadrangular with a pronounced depression that divides the anterior and posterior lobes, and shorth copulatory ducts (Figs. 22D and 22E).

**Male** (CNAN-AR-T01865, from Los Tuxtlas Biological Station, Veracruz province, Mexico). Total length 26.87. Carapace 14.11 long and 11.04 wide. AME 0.54, ALE 0.42, PME 0.53, PLE 0.54. Sternum 6.16 long and 5.42 wide, labium 1.90 long and 1.70, endites 3.23 long and 1.89 wide. Leg measurements: I: femur 16.87, patella 5.67, tibia 18.67, metatarsus 16.84, tarsus 6.84, total 64.89; II: 16.61, 6.45, 16.88, 15.85, 6.47, total 62.26; III: 14.68, 5.34, 13.85, 13.83, 5.20, total 52.90; IV: 17.82, 5.86, 18.30, 11.81, 6.2, total 31.58. Leg spination: leg I tibia v2-2-2-2-2, p1-1-0, r1-1-0, metatarsus v2-2-2; d1-0-1, p1-1-0, r1-1-0; II-tibia v2-2-2-2-2, d0-1-0, p1-1-0, r1-1-0, metatarsus v2-2-2, p1-1-0, r1-1-0; III-tibia v2-2-2, d1-1-1, p1-0-1, r1-0-1, metatarsus v2-2-2, d0-1-0, p1-1-1, r1-1-1; IV-tibia v2-2-2, d1-1-1, p1-01, r1-0-1, metatarsus modified. Palp: as in genus description.

**Female** (CNAN-AR-T01866, from Los Tuxtlas Biological Station, Veracruz province, Mexico). Total length 24.78. Carapace 11.28 long and 8.89 wide. AME 0.50, ALE 0.35, PME 0.50, PLE 0.50. Sternum 4.52 long and 4.32 wide, labium 1.23 long and 2.54 wide, endites 2.24 long and 1.70 wide. Leg measurements: I: femur 11.61, patella 4.85, tibia 12.32, metatarsus 9.89, tarsus 4.06, total 42.73; II: 11.09, 4.76, 11.13, 9.02, 3.71, total 39.71; III: 12.40, 4.08, 8.70, 8.35, 3.04, total 36.57; IV: 12.23, 4.32, 11.28, 12.28, 4.10, total 44.21. Leg spination: tibia I-II v2-2-2-2-2, metatarsus I-II v2-2-2-2-2, tibia III v-2-2-2, d1-1-1, p1-0-1, r1-0-1, metatarsus v2-2-2, d0-1-0, p1-1-2, r1-1-2; IV-tibia v2-2-2, d1-1-1, 1-0-1, p1-0-1, metatarsus v-2-2-2, d0-1-0, p1-1-2, r1-1-2. Epigynum: as in generic description. Epigynum: as in genus description.

**Distribution.** Lowland and premontane ecosystem of Mexico (Fig. 27).

## DISCUSSION

This study presents a molecular phylogeny of *Kiekie* including 15 of the 16 currently described species. Our phylogenetic analyses support the monophyly of *Kiekie* and reveal a new genus from Mexico as its sister lineage. The monophyly of *Kiekie* is supported by three morphological synapomorphies: an elongated embolus with a laminar process (Figs. 23A–23F, 24A–24F), conductor resembling an open fan (Figs. 25A–25F, 26A–26F), and locking lobes located at posterior and sometimes in retro-posterior side (Figs. 25B, 26C), instead of posterior prolateral as most ctenines. The sister relationship of *Eldivo* and *Kiekie* is supported by the presence of a modified IV metatarsus in males (Fig. 4C), although this character has evolved convergently several times in South American ctenines. The *Eldivo* monophyly is supported by one male morphological synapomorphy: a laminar fold in the internal side of the embolus.

Most of the intrageneric relationships of *Kieke* are well supported by molecular data, and some of them also by morphological characters. The first clade comprises the species *K. curvipes*, *K. verbena* and *K. montanense* shared a prominent tegular process. The second clade contains the larger species of the genus: *K. sinuatipes*, *K. barrantesi* sp.nov, *K. sarapiqui*, *K. laselva* sp.nov, *K. valerioi* sp.nov, *K. panamensis*, *K. griswoldi* and *K. lascruces*. Although the second clade is recovered with low support, it also presents several synapomorphies, providing more support to its monophyly: elongate and curved copulatory ducts that project in the anterior area of the epigynum, and large copulatory openings. In addition, males of the second lineage have longer flagelliform embolus compared to the remaining species. Within this clade, the sister relationship between *K. sinuatipes* and *K. barrantesi* sp.nov is supported by the copulatory ducts having a 360° turn that completely covers the anterior side of the spermathecal, while in the remaining species of this clade the copulatory ducts fold over itself before it enters the spermathecae.

The highest species diversity in *Kiekie* occurs in the montane ecosystems of Costa Rica, followed by the lowland rainforest of the Pacific side (Limón Basin). This diversity arose after a dispersal event from the tropical region of North America to Lower Central America (Costa Rica) during the Late Miocene (10-7ma). The northern origin of *Kiekie* and subsequently colonization of Lower Central and South America is supported not only by the early divergent North American species *K. garifuna* and the sister genus *Eldivo*, but also by a more basal lineage of North American species of the genera *Ctenus* and *Leptoctenus* (Peck, 1981). Therefore, the northern origin of *Kiekie* and subsequent dispersal and diversification in Lower Central America is well supported. Interestingly, the species diversity of *Kiekie* starts to decrease toward the south and in the Trans-Andean region of South America is only represented by two species, being *K. panamensis* the most common species found in Colombia and Ecuador. The high diversity of *Kiekie* in Lower Central America which can reach up to five species in one locality of tropical rainforest is replaced in South America by another ecological equivalent ctenid, *Spinoctenus* (*Hazzi et al., 2018*).

The geological evolution of the Costa Rica forearc has a complex history related to subduction along the Middle America Trench (*Porras et al., 2021*). Geochronological,

geochemical, and petrological data from the Talamanca Cordillera tracks the key turning point (12–8 Ma) from the evolution of an oceanic arc depleted in incompatible elements to a juvenile continent (*Gazel et al., 2019*). The first event of dispersal from Tropical North America during the late Miocene temporally corresponds with these dates in which Costa Rica started becoming a young continental land.

Although there is no information about the different elevation stages of the mountain ranges in Costa Rica at different time periods, there are contractional deformation periods in Cordillera Central and Talamanca from 11 to 5 ma (*Mescua et al., 2017*; *Porras et al., 2021*). In the case of Cordillera Central the period of deformation is from 10 to 5 ma, which it could be enough time to allow the formation of mild elevation mountains (*Porras et al., 2021*). In Talamanca Cordillera, the deformation in the Fila Costeña was probably linked to deformation and uplift of this Cordillera in the middle to late Miocene (*Mescua et al., 2017*). These deformations are also related with a change of the chemistry of the volcanic rocks which could indicate mountain uplift (*Porras et al., 2021*). The four independently events of Colonization from lowland ecosystems to the montane ecosystems of Cordillera Central and Talamanca during the Late Miocene (9–4 ma) temporally match the geological time assumed in which Central and Talamanca Cordilleras started uplifting.

The divergence times of montane species of *Kiekie* also correspond with the time of diversification of montane palm-pitviper snakes (*Bothriechis*) in Costa Rica which occurred during the Late Miocene (*Mason et al., 2019*). These events of deformation and uplift also coincided with the deposition of the Limón basin (Pacific side of Costa Rica) which contributed to the final closure of the Isthmus, and possibly causing the speciation of the three lowland endemic species of this region: *K. laselva* sp.nov, *K. tirimbina* sp.nov and *K. sarapiqui*, around 7–4 ma. The diversification of these lowland species also coincides to some extent with the origin times of *Brachyrhaphis* fishes (Poecilidae) in the Pacific Region of Central America (*Ingley et al., 2015*).

## CONCLUSIONS

This study increased the number of known species of *Kiekie* from 11 to 16 and documented a new genus, *Eldivo* which is the sister lineage of *Kiekie*. Most of the diversity and endemism of *Kiekie* is located in the montane ecosystems of Costa Rica followed by the lowland rainforest of the Pacific side (Limón Basin). The high abundance and specificity of *Kiekie* species in the leaf litter within Costa Rican forest ecosystems suggests their potential as valuable models for ecological and conservation biology studies (*Hazzi et al., 2020*). *Kiekie* originated in the North America Tropical region and dispersed to Lower Central America where it started diversifying during the Late Miocene. In Lower Central America, *Kiekie* colonized independently several times the montane ecosystems corresponding to periods of uplifting of Talamanca and Central Cordilleras. Then, during the Pliocene *Kiekie* dispersed to South America as attested by the species *Kiekie panamensis*.

## MUSEUM ABBREVIATIONS

The material examined and/or collected belongs to the following museums:

**MUSENUV** Museo Entomológico de la Universidad del Valle, Cali, Colombia (J. Cabra)

**MZUCR** Museo de Zoología, Escuela de Biología, Universidad de Costa Rica (G. Barrantes)

**MNCR** Museo Nacional de Costa Rica (M. Sánchez-Ocampo, includes former Instituto Nacional de Biodiversidad de Costa Rica (INBio) collection)

**MCZ** Museum of Comparative Zoology, Harvard University, Cambridge, USA (G. Giribet)

**CNAN-AR** National Arachnid Collection at Universidad Nacional Autónoma de México, Mexico (O. Francke and E. González).

## ACKNOWLEDGEMENTS

We are grateful to Gilbert Barrentes (MZUCR), Marcela Sanchez Ocampo (MNCR), Oscar Francke and Edmundo Gonzalez (CNAN) and Jimmy Cabra (MUSENUV) for making available some of the specimens for this study. We also want to thank to La Tirimbina Rainforest Center for their great hospitality and help during fieldwork. N. Hazzi is deeply in debt with Gilbert Barrantes, Natalia Conejo, Bernal Rodríguez Herrera, Ronald Cordero, Juan Bernal and Tomas A. Rios for fieldwork logistics and travel. N. Hazzi also want to thanks to William Eberhard and Mary West-Eberhard for their hospitality. We thank José Mescua for providing us valuable literature about the geology and mountain uplift of Costa Rica. We thank Ivan Magalhaes, Diana Silva and a anonymous reviewer for their suggestions and corrections on a previous version of the manuscript.

### Funding

This study was supported by the Department of Biological Sciences of The George Washington University, the Harlan Fellow-ship, the Explorers Club Washington, DC group and The Early Career Grants of National Geographic, the Ernst Mayr TravelGrants (MCZ, Harvard University) for their travel sup-port to examine the ctenid specimens of the Museo de Zoologia (MZUC), Universidad de Costa Rica. Nicolas Hazzi was supported by a Fulbright-Colciencias scholarship and the Office of Graduate Student Assistantships and Fellowships of The George Washington University. Additional support was provided by a US National Science Foundation grants (DEB 1754289, DEB 1754289) to Gustavo Hormiga. The funders had no role in study design, data collection and analysis, decision to publish, or preparation of the manuscript.

### Grant Disclosures

The following grant information was disclosed by the authors:

Department of Biological Sciences of The George Washington University.

DC group and The Early Career Grants of National Geographic, the Ernst Mayr

TravelGrants (MCZ, Harvard University).
Museo de Zoologia (MZUC), Universidad de Costa Rica.
Fulbright-Colciencias scholarship and the Office of Graduate Student Assistantships and Fellowships of The George Washington University.
US National Science Foundation grants: DEB 1754289, DEB 1754289.

## Competing Interests

The authors declare that they have no competing interests. Nicolas Hazzi is a co-founder of Fundacion Ecotonos.

## Author Contributions

- Nicolas Hazzi conceived and designed the experiments, performed the experiments, analyzed the data, prepared figures and/or tables, authored or reviewed drafts of the article, and approved the final draft.
- Gustavo Hormiga conceived and designed the experiments, performed the experiments, prepared figures and/or tables, authored or reviewed drafts of the article, and approved the final draft.

## DNA Deposition

The following information was supplied regarding the deposition of DNA sequences:

The sequences are available at NCBI: OR834109, OR834126, OR882106, OR834091, OR834155, OR837527, OR834093, OR837526, OR834092, OR834111, OR882093, OR834147, OR834094, OR834112, OR837508, OR834070, OR834098, OR834115, OR882097, OR834077, OR834116, OR837512, OR834078, OR834096, OR834132, OR837510, OR882094, OR834073, OR834148, OR834097, OR834133, OR837511, OR834074, OR834149, OR834104, OR834142, OR837522, OR882104, OR834086, OR834153, OR834108, OR834125, OR837525, OR882105, OR834090, OR834154, OR834138, OR834119, OR837517, OR834083, OR834151, OR834114, OR834150, OR882095, OR834075, OR882096, OR834076, OR837521, OR834095, OR834131, OR834113, OR837509, OR882103, OR834071, OR834072, OR834102, OR834140, OR834120, OR837519, OR882101, OR834085, OR834152, OR834103, OR834141, OR834121, OR837520, OR834101, OR834137, OR837516, OR882107, OR834082, OR834105, OR834143, OR834122, OR882100, OR834087, OR834106, OR834144, OR834123, OR837523, OR834088, OR834107, OR834145, OR834124, OR837524, OR882102, OR834089, OP214431, OR837518, OR834084, OR834134, OR837513, OR834079, OR834099, OR834135, OR834117, OR837514, OR882098, OR834080, OR834100, OR834136, OR834118, OR837515, OR882099, OR834081.

## Data Availability

Concatenated alignment of the DNA sequences and sequence partition is available in the Supplemental Files.

## New Species Registration

The following information was supplied regarding the registration of a newly described species:

Publication LSID: urn:lsid:zoobank.org:pub:A693D4C7-C4DD-4F69-9D2A-DDD62527B4CF

Eldivo genus LSID: urn:lsid:zoobank.org:act:F9310193-9241-4048-91F9-1D81A198066F

Eldivo tuxlas species LSID: urn:lsid:zoobank.org:act:918F4530-36A6-453D-8F24-EB8DE451B3C4

Kiekie bernali species LSID: urn:lsid:zoobank.org:act:3F86BEFF-2211-478F-8159-0D80D164EC08

Kiekie lamuerte species LSID: urn:lsid:zoobank.org:act:9533BADD-C0A7-4334-B77C-6FC28914F423

Kiekie laselva species LSID: urn:lsid:zoobank.org:act:E3AAC76A-48AF-4601-ADEF-F559840FE6C3

Kiekie tirimbina species LSID: urn:lsid:zoobank.org:act:D0A1697B-F009-45DE-8933-13EADB2ACFE0

Kiekie valeroi species LSID: urn:lsid:zoobank.org:act:ADDA878A-CB6A-4097-AA5E-4D44A9F502C4.

## Supplemental Information

Supplemental information for this article can be found online at http://dx.doi.org/10.7717/peerj.17242#supplemental-information.

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
