# Peer review of "Systematics, distribution patterns and historical biogeography of the Central America wandering spider genus Kiekie Polotow & Brescovit, 2018 (Araneae: Ctenidae)"

_PeerJ, doi:10.7717/peerj.17242_

## Round 0.1 · original submission · Minor Revisions

I have read the manuscript and the reviews. I also agree that this paper is interesting and adds a lot of information to this group of spiders.
Please read carefully the revisions, I am sure the manuscript can only improve.

·

Basic reporting

There are some minor grammar issues to be improved, for example, be consistent in the use of "et al." with italics. Please, check throughout the text and change to K. sinuatipes, there are some misspellings in this species name. You need to check also the literature references, at least one is incomplete.

Experimental design

No comment

Validity of the findings

No comment

Additional comments

Some comments added to the text

Reviewer 2 ·

Basic reporting

The manuscript addresses the review of the genus Kiekie that includes taxonomy, systematics and biogeography. It is an excellent example of an integrative study that provides relevant results for the identification of species, their phylogenetic relationships and diversification in space and time. There is an important work of reviewing material deposited in different institutions and field samplings carried out by the authors that allowed molecular analyzes.
The study provided a significant number of new species of the genus Kiekie, increased distribution data of already known species, described new sexes and therefore provided new diagnoses. Based on molecular studies, a new genus was also described. The images are clear and allow to recognize the diagnostic structures of the species. The biogeographic analysis shows the diversification of the genus Kiekie as well as the areas of greatest richness and endemism. This result is very valuable since it generates new inputs for conservation plans in mountain and lowland areas of Central America.
The English is correct.
In the Introduction, update the number of species described today in this genus and include new citations in References. This section needs revision.
Figures and Supplementary files are relevant to the content of the article.
The comments that I made are included in the attached text and most can be considered minor changes. Considering the number of species in the Kiekie genus, I suggest to the authors that a species recognition key would be a very interesting complement to include.

Experimental design

The manuscript agrees with the scope of this journal.
Research questions are correctly stated and the knowledge gap was higly resolve.
It is an exhaustive study with revision of material in different institutions and several collecting surveys.
Methods used are clearly described, replicated and sufficient for the study developed.
The integrative approach with different methodologies used constitutes a relevant contribution to carry out the study.

Validity of the findings

The results are relevante. They improved the knowledge about the taxonomy of the genus Kiekie with the description of new species and the description of unknown sexes for the species already described.
The conclusions are well stated and linked to the original questions of the study.

Additional comments

No comment

Annotated reviews are not available for download in order to protect the identity of reviewers who chose to remain anonymous.

·

Basic reporting

This manuscript presents a phylogenetic and taxonomic revision of Kiekie, a group of wandering spiders especially diverse in Mesoamerican highlands. It is excellently designed and conducted. Both the systematic and biogeographic questions are addressed satisfactorily, the work is well written and beautifully illustrated. My only major comment regards the diagnoses and the availability of the phylogenetic matrices.

- Some of the diagnoses need to be slightly re-worked: for instance, “Diagnosis. Males resemble to those of K. panamensis and K. valeroi, but can be distinguished from them by…”. It would be more helpful if the authors could indicate exactly what makes these species resemble each other. This may be self-evident to a Ctenidae taxonomist, but it would be important that it would be more explicit to non-specialists. Please consider that not only other taxonomists will use your work, but also ecologists, who may never have looked at a ctenid palp/epigyne before. Please revise this throughout the manuscript.
- Please provide a diagnosis for females of Eldivo tuxtlas; I understand you cannot propose a generic diagnosis that works to separate females of the different genera, but surely this species has a particular genital shape that allows recognizing it.
- Please provide the phenotypic and molecular matrices (raw and processed alignments) as supplementary material.

The rest of my suggestions are cosmetic or indicate corrections to small mistakes, see below. I commend the authors on this excellent piece of research.

- Some references are missing (e.g. Jaramillo 2019). Please revise all of them.
- Please indicate if outgroups were pruned before the RASP analysis, and what was the rationale for the pruning.
- Please show all supports in the tree, not only those above 0.95
- Some portions of the text, including the genus composition, still lack Omelko’s (2023) species and thus cites only 16, not 18, species. Please revise.
- Perhaps after each species (including previously described ones), can you please add a “Distribution” section indicating where the species occurs, and refer to the maps?
- The legends and symbols in the maps are small and difficult to see. Is it possible to make them larger?
- Please correct Kiekie curvipes Polotow & Brescovit, 2018 to Kiekie curvipes (Keyserling, 1881)
- The format of coordinates is not consistent: for example, some read 10.4498N, -83.4730W, other read 10.4150N, -84.1210. I suggest using either N/S and E/W, or positive/negative values, but not both; the former is preferable. Please use the same format throughout the manuscript.
- Please correct the spelling of K. valeroi, as the patronym is incorrectly formed. If it is a patronym after Valerio, it should be written valerioi by taking Valerio and adding the genitive suffix -i; alternatively, because Valerio appears like a Latinized name (even though it is not – it should be Valerius), it is possible to treat it like a Latin name, take its root (valeri-) and make it valerii. Please refer to this quite informative page for more details: https://entnemdept.ufl.edu/frank/kiss/kiss24.htm
- Please correct sinautipes to sinuatipes throughout the paper.

See additional minor corrections appended directly to the manuscript.

Experimental design

The design is adequate to the questions.

Validity of the findings

Please provide the phenotypic and molecular matrices (raw and processed alignments) as supplementary material.

---

## Round 0.2 · accepted · Accept

I have read the revised version of this manuscript and the authors paid close attention to all comments made by the three reviewers. I believe the manuscript is in very good shape now and can be accepted for publication